



# Applicability and consequences of the integration of alternative models for $CO_2$ transfer velocity into a process-based lake model

Petri Kiuru[1,2], Anne Ojala[3,4,5], Ivan Mammarella[6], Jouni Heiskanen[6,7], Kukka-Maaria Erkkilä[6], Heli Miettinen[8], Timo Vesala[4,6], and Timo Huttula[1]

[1]Finnish Environment Institute, Freshwater Centre, Survontie 9A, FI-40500 Jyväskylä, Finland
[2]University of Jyväskylä, Department of Physics, P.O. Box 35, FI-40014 University of Jyväskylä, Finland
[3]Faculty of Biological and Environmental Sciences, Ecosystems and Environment Research Programme, University of Helsinki, Niemenkatu 73, FI-15140 Lahti, Finland
[4]Institute for Atmospheric and Earth System Research/Forest Sciences, Faculty of Agriculture and Forestry, University of Helsinki, P.O. Box 27, FI-00014 Helsinki, Finland
[5]Faculty of Biological and Environmental Sciences, Helsinki Institute of Sustainability Science, University of Helsinki, Finland
[6]Institute for Atmospheric and Earth System Research/Physics, Faculty of Science, University of Helsinki, P.O. Box 68, FI-00014 Helsinki, Finland
[7]ICOS ERIC Head Office, Erik Palménin aukio 1, FI-00560 Helsinki, Finland
[8]Faculty of Biological and Environmental Sciences, University of Helsinki, P.O. Box 65, FI-00014 Helsinki, Finland

**Correspondence:** Petri Kiuru (petri.kiuru81@gmail.com)

**Abstract.** Freshwater lakes are important in carbon cycling especially in the boreal zone, where many lakes are supersaturated with the greenhouse gas carbon dioxide ($CO_2$) and emit it to the atmosphere, thus ventilating carbon originally fixed by the terrestrial system. The exchange of $CO_2$ between water and the atmosphere is commonly estimated using simple wind-based parameterizations or models of gas transfer velocity ($k$). More complex surface renewal models, however, have been shown to yield more correct estimates of $k$ in comparison with direct $CO_2$ flux measurements. We incorporated four gas exchange models with different complexity into a vertical process-based physicobiochemical lake model MyLake C and assessed the performance and applicability of the alternative lake model versions to simulate air–water $CO_2$ fluxes over a small boreal lake. None of the incorporated gas exchange models significantly outperformed the other models in the simulations in comparison to the measured near-surface $CO_2$ concentrations or respective air–water $CO_2$ fluxes calculated directly with the gas exchange models using measurement data as input. The use of more complex gas exchange models in the simulation, on the contrary, led to difficulties in obtaining sufficient gain of $CO_2$ in the water column and thus resulted in lower $CO_2$ fluxes and water column $CO_2$ concentrations compared to the respective measurement-based values. Inclusion of sophisticated and more correct models for air–water $CO_2$ exchange in process-based lake models is crucial in efforts to properly assess lacustrine carbon budgets through model simulations both in single lakes and on a larger scale. However, finding higher estimates for both the internal and the external sources of inorganic carbon in boreal lakes is important if the improved knowledge of the magnitude of $CO_2$ evasion from lakes is included in future studies on lake carbon budgets.





# 1 Introduction

The majority of inland waters, especially in the boreal zone, are found to be supersaturated with carbon dioxide ($CO_2$) with concentrations that can exceed the equilibrium concentration by several times and are therefore net sources of carbon to the atmosphere (Cole et al., 1994; Algesten et al., 2014). The contribution of lakes to the global carbon budget is recognized to

be substantial in comparison to the role of marine and terrestrial ecosystems as global carbon sinks, but global quantitative estimates show significant variation (Cole et al., 2007; Battin et al., 2009; Tranvik et al., 2009). Atmospheric $CO_2$ exchange between lakes and the atmosphere is one of the key processes needed to be determined in constructing carbon budgets of lakes and in evaluating the role of lakes in global carbon cycling.

The exchange of weakly soluble gases, like $CO_2$ and oxygen, across the air–water interface is often modeled as a boundary-

layer process in which the gas flux is proportional to the gas concentration gradient at the interface. The proportionality factor $k$ is known as the gas transfer velocity. In many long-used models for the gas transfer velocity, or gas exchange models, $k$ is parameterized by wind speed alone (Wanninkhof 1992, Cole and Caraco 1998). However, direct measurements of air–water $CO_2$ exchange using the eddy covariance (EC) method (Jonsson et al., 2008; MacIntyre et al., 2010; Heiskanen et al., 2014) have resulted in higher estimates of $k$ compared to wind-based gas exchange models. For weakly soluble gases, $k$ depends

mainly upon turbulence in near-surface water (Banerjee, 2007), which is not generated merely by wind. Near-surface turbulence is initiated predominantly by wind shear and negative buoyancy flux related to thermal convection induced by surface heat loss (Imberger, 1985). Buoyancy flux is relatively more important in small, wind-sheltered lakes (Read et al., 2012). Turbulence-driven gas exchange models have been shown to be well in accordance with in situ measurements of $k$ (e.g., Zappa et al., 2007; Vachon et al., 2010).

In surface renewal models, $k$ is calculated as a function of the turbulent kinetic energy dissipation rate $\varepsilon$, which provides an indication of the intensity of near-surface turbulence (MacIntyre et al., 1995). Kinetic energy dissipation can be due to viscous and thermal processes, and $\varepsilon$ is thus dependent on wind shear and convective heat flux (Lombardo and Gregg, 1989). Wind shear is characterized by the wind-induced water-side friction velocity. The water-side friction velocity can be estimated from the atmospheric friction velocity, which can be measured directly (Mammarella et al., 2015) or calculated by bulk formulas

using meteorological variables (Fairall et al., 1996). Heat-induced turbulence is generated if the surface heat flux is negative, that is, directed out of the lake. If measurements of the components of surface heat flux are not available, they can also be estimated using bulk formulas (Fairall et al., 1996).

Global estimates of carbon emissions from lakes often use conservative estimates of $CO_2$ fluxes or models that yield potentially underestimated values for $k$ leading to low estimates of $CO_2$ fluxes (e.g., Cole et al., 2007; Raymond et al., 2013). Thus,

revised estimates of lacustrine $CO_2$ emissions will require higher terrestrial ecosystem production to close the global carbon balance (Battin et al., 2009). Many studies concerning modeling of lake carbon balance (e.g., Bade et al., 2004; McDonald et al., 2013) or determination of lake carbon budgets (e.g., Sobek et al., 2006; Stets et al., 2009; Chmiel et al., 2016) also use simple wind-based models for $k$. Potential subsequent underestimates in carbon efflux may have consequences for the interpretation of carbon budgets in single lakes (Dugan et al., 2016). A higher efflux may result in a re-evaluation of the amount of





net ecosystem production in lakes or it can mean that external carbon sources are inadequately accounted for in lake carbon budgets.

The efflux of $CO_2$ from a lake is sustained mainly by in-lake $CO_2$ production through bacterial or photochemical degradation of organic matter in water column or in sediment. Widely across the boreal zone, the importance of the degradation of
allochthonous organic matter as an inorganic carbon source in lakes is conspicuous (Jonsson et al., 2001; Sobek et al., 2003). Also the direct loading of terrestrially produced dissolved inorganic carbon (DIC) through surface water and groundwater inflows may lead to high $CO_2$ concentrations in some lakes (Maberly et al., 2013; Weyhenmeyer et al., 2015; Einarsdóttir et al., 2017).

In this study, we evaluated the performance of different gas exchange models in the simulation of air–water $CO_2$ flux in
a boreal lake with a process-based lake model and the adaptability of the lake model application to different $CO_2$ losses via efflux. We also calculated $CO_2$ budgets for the epilimnion of the lake during summer stratification on the basis of the simulation results and assessed the relative importance of different biogeochemical processes on the epilimnetic $CO_2$ conditions. We incorporated four alternative gas exchange models into a vertical process-based physico-biogeochemical lake model for the simulation of year-round profiles of water temperature and $CO_2$ concentrations with a daily time step. We then applied the lake
model to a humic boreal lake located in southern Finland for the period 2013–2014, calibrating each of the resultant alternative lake model versions against high-frequency water column $CO_2$ concentration measurements. We compared the simulated gas transfer velocities and air–water $CO_2$ fluxes with those calculated with the gas exchange models on the basis of measurement data. The aims of our study are (i) to assess the applicability of gas exchange models of different complexity to a process-based lake model with a daily time step and (ii) to assess the implications of higher $CO_2$ efflux estimates for the lake carbon budget.

## 2 Materials and methods

### 2.1 Modeling approach

#### 2.1.1 Lake model MyLake C

We used an application of a one-dimensional process-based lake model MyLake C (Kiuru et al., 2018) for the simulation of the vertical distributions of water column temperature and $CO_2$ concentration and air–water $CO_2$ flux in the study lake.
MyLake C simulates inorganic and organic carbon cycling in a lake, taking into account terrestrial carbon loading, air–water exchange of $CO_2$, and changes in water column pH. However, groundwater exchange and changes in water level due to rainfall or evaporation are excluded. The model operates on a daily time step, and the vertical grid length can be defined by the user. The model is based on a lake model MyLake v.1.2 (Saloranta and Andersen, 2007), which simulates lake thermal structure, seasonal ice and snow cover, and phosphorus-phytoplankton dynamics. In the model, vertical heat and mass diffusion are
calculated with a diffusion equation using a vertical turbulent diffusion coefficient derived from the buoyancy frequency and parameterized by lake surface area by default. Settling of particulate substances is also taken into account in the equation. In addition, convective and wind-induced water column mixing processes are included. As an exception to the daily time step,





heat exchange between the water column and the atmosphere is calculated separately for daytime and nighttime. MyLake v.1.2 and its various extensions have been used in studies on stratification and lake ice cover (e.g., Saloranta et al., 2009; Dibike et al., 2012; Gebre et al., 2014), total phosphorus concentration and phytoplankton biomass (e.g., Romarheim et al., 2015; Couture et al., 2018), dissolved organic carbon (DOC) concentration (Holmberg et al., 2014; de Wit et al., 2018), and dissolved oxygen
(DO) conditions (Couture et al., 2015).

MyLake C has been designed to include only the most substantial physical, chemical, and biological processes related to carbon cycling in a well-balanced and robust way. $CO_2$ is produced in the lake through organic carbon degradation both within the water column and in the sediment and through phytoplankton respiration. Inorganic carbon production is coupled to DO consumption, and vice versa. A division is made between readily degradable, phytoplankton-originated autochthonous
particulate organic carbon (POC) and more refractory allochthonous POC. The model includes also the sedimentation, the resuspension and the permanent burial of POC. Correspondingly, DOC is classified into three compound classes with different bacterial degradabilities. A separate submodule (Holmberg et al., 2014) calculates the conversion of DOC into an inorganic form via bacterial and photochemical degradation. The meteorological model forcing includes daily global radiation, cloud cover fraction, atmospheric temperature, relative humidity, atmospheric pressure, wind speed at $10\,\mathrm{m}$ height, and precipitation.
Hydrological forcing data include daily inflow volumes, inflow temperatures, inflow pH, and the inflow concentrations of modeled substances, including DOC, POC, and DIC. Complete data requirements are presented and model structure and applied equations are described in detail in Kiuru et al. (2018).

MyLake uses the Air-Sea Toolbox (Air-Sea, 1999) based on the parameterizations and algorithms in Fairall et al. (1996) for calculation of surface wind stress and the components of surface heat flux. The sensible heat flux $Q_\mathrm{H}$, the latent heat flux $Q_\mathrm{L}$,
and the wind shear stress $\tau$ are obtained from aerodynamic bulk formulas of the form

$$Q_\mathrm{H} = \rho_\mathrm{a} c_{pa} C_\mathrm{h} U (T_\mathrm{a} - T_\mathrm{s}) \tag{1}$$

$$Q_\mathrm{L} = \rho_\mathrm{a} L_\mathrm{e} C_\mathrm{l} U (q_\mathrm{a} - q_\mathrm{s}) \tag{2}$$

$$\tau = \rho_\mathrm{a} C_\mathrm{d} U^2, \tag{3}$$

where $\rho_\mathrm{a}$ is the air density, $c_{pa}$ is the specific heat capacity of air, $C_\mathrm{h}$ and $C_\mathrm{l}$ are the transfer coefficients of sensible and
latent heat, respectively, $C_\mathrm{d}$ is the drag coefficient, $U$ is wind speed, $T_\mathrm{a}$ is air temperature, $T_\mathrm{s}$ is water surface temperature, $L_\mathrm{e}$ is the latent heat of evaporation of water, $q_\mathrm{a}$ is the specific humidity, and $q_\mathrm{s}$ is the saturation specific humidity at the water surface temperature. No wind sheltering effect on $U$ is applied in the calculation of surface wind stress and surface heat flux components.

The flux of $CO_2$ between water and the atmosphere, $F_{CO2}$, given in units of $\mathrm{mg\,m^{-2}\,d^{-1}}$ in MyLake C, is calculated as
the product of the $CO_2$ concentration gradient between the surface water and the atmosphere and the gas transfer velocity $k$ $(\mathrm{m\,s^{-1}})$ as

$$F_{CO2} = \alpha k (C_\mathrm{w} - C_\mathrm{eq}), \tag{4}$$





where $C_{\mathrm{w}}$ ($\mathrm{mg\,m^{-3}}$) is the $CO_2$ concentration in the topmost model grid layer representing the water surface, $C_{\mathrm{eq}}$ ($\mathrm{mg\,m^{-3}}$) is the equilibrium concentration of $CO_2$, that is, the water column $CO_2$ concentration in the state of equilibrium with the overlying atmosphere, and $\alpha$ is the chemical enhancement factor applicable for reactive gases, such as $CO_2$. Fluxes from water to the atmosphere are thus defined to be positive. If a lake is nonalkaline, $\alpha$ can assumed to be 1 (Cole and Caraco, 1998), which is also the default value in the model. The equilibrium concentration is calculated by Henry's law as

$$C_{\mathrm{eq}} = K_{\mathrm{H}}\chi p_{\mathrm{a}}, \tag{5}$$

where $K_{\mathrm{H}}$ is the Henry's law constant for the gas at surface water temperature, $\chi$ is the mole fraction of the gas in the atmosphere, and $p_{\mathrm{a}}$ is the atmospheric pressure. The temperature dependence of the solubility of $CO_2$ is calculated according to Weiss (1974).

### 2.1.2 Gas exchange models

In MyLake C, the calculation of $k$ is performed using the widely applied experimental wind-based regression formula by Cole and Caraco (1998), which gives the gas transfer velocity in units of $\mathrm{cm\,h^{-1}}$ as

$$k_{\mathrm{CC}} = (2.07 + 0.215U_{10}^{1.7})\Big(\frac{Sc}{600}\Big)^{-0.5}, \tag{6}$$

where $U_{10}$ ($\mathrm{m\,s^{-1}}$) is the wind speed at 10 m and $Sc$ is the temperature-dependent Schmidt number determined for surface water conditions using the polynomial fit in Wanninkhof (1992). The approximation $U_{10}/U_{1.5} = 1.22$, where $U_{1.5}$ is the wind speed measured at 1.5 m, is used in the calculations. In this study, we incorporated the models for $k$ by MacIntyre et al. (2010), Heiskanen et al. (2014), and Tedford et al. (2014) into MyLake C as alternatives to the default model by Cole and Caraco (1998).

In a surface renewal model by Tedford et al. (2014), the gas transfer velocity $k_{\mathrm{TE}}$ ($\mathrm{m\,s^{-1}}$) is parameterized by the total turbulent kinetic energy dissipation rate $\varepsilon_{\mathrm{TE}}$ as

$$k_{\mathrm{TE}} = c(\nu\varepsilon_{\mathrm{TE}})^{0.25}Sc^{-0.5}. \tag{7}$$

where $c$ is a dimensionless constant and $\nu$ is the kinematic viscosity of water. In the model, both wind-induced stress and heat-induced convection generate turbulence near the lake surface during cooling, but wind is the only factor responsible for the turbulence during heating. The total turbulent kinetic energy dissipation rate is determined in terms of shear production $\varepsilon_{\mathrm{s}} = u_{*\mathrm{w}}^3/\kappa z'$, where $u_{*\mathrm{w}}$ is the wind-induced water-side friction velocity, $\kappa = 0.4$ is the von Kármán constant, and $z'$ is a reference depth, and convective turbulence production $\varepsilon_{\mathrm{c}}$ equaling the buoyancy flux $\beta$, as

$$\varepsilon_{\mathrm{TE}} = \begin{cases} 0.56\varepsilon_{\mathrm{s}} + 0.77\,|\varepsilon_{\mathrm{c}}| & \text{if } \beta < 0, \\ 0.6\varepsilon_{\mathrm{s}} & \text{if } \beta \geq 0, \end{cases} \tag{8}$$

In this study, the constants are defined as $c = 0.5$ and $z' = 0.15$ m as in Erkkilä et al. (2018). The wind-induced water friction velocity $u_{*\mathrm{w}}$ is calculated from the atmospheric friction velocity $u_{*\mathrm{a}} = (\tau/\rho_{\mathrm{a}})^{0.5}$ as in MacIntyre et al. (1995)

$$u_{*\mathrm{w}} = u_{*\mathrm{a}}\Big(\frac{\rho_{\mathrm{a}}}{\rho_{\mathrm{w}}}\Big)^{0.5}, \tag{9}$$





where $\rho_\mathrm{w}$ is the density of water.

The buoyancy flux is defined as in Imberger (1985)

$$\beta = \frac{g\alpha_\mathrm{w}Q_\mathrm{eff}}{\rho_\mathrm{w}c_{pw}}, \tag{10}$$

where $g$ is the gravitational acceleration, $\alpha_\mathrm{w}$ is the thermal expansion coefficient of water, $Q_\mathrm{eff}$ is the effective heat flux, and
$c_{pw}$ is the specific heat capacity of water. The effective heat flux is defined as

$$Q_\mathrm{eff} = Q_\mathrm{S} + Q_\mathrm{SW}(0) + Q_\mathrm{SW}(z_\mathrm{AML}) - \frac{2}{z_\mathrm{AML}} \int\limits_{0}^{z_\mathrm{AML}} Q_\mathrm{SW}(z)\mathrm{d}z, \tag{11}$$

where $Q_\mathrm{S} = Q_\mathrm{H} + Q_\mathrm{L} + Q_\mathrm{LW}$ is the net surface heat flux, $Q_\mathrm{LW}$ is net longwave radiation, $Q_\mathrm{SW}$ is shortwave radiation, and
$z_\mathrm{AML}$ is the depth of the actively mixing layer (AML). The last three terms in the equation represent the fraction of shortwave
radiation that is trapped within the AML, denoted as $Q_\mathrm{SW,AML}$. The attenuation of shortwave radiation at depth $z$ in the water
column is calculated using the Beer–Lambert law

$$Q_\mathrm{SW}(z) = Q_\mathrm{SW}(0)\mathrm{e}^{-K_\mathrm{L}z}, \tag{12}$$

where $K_\mathrm{L}$ is the total attenuation coefficient of shortwave radiation. The AML is defined as the near-surface layer in which
the water column temperature is within $0.02\,^\circ\mathrm{C}$ of the temperature at the air–water interface (MacIntyre et al., 2001). All heat
fluxes from the atmosphere into the lake are defined positive. Thus, the buoyancy flux is positive when the near-surface water
is heating and negative under cooling conditions.

In a boundary-layer model developed by Heiskanen et al. (2014), near-surface turbulence is parameterized through wind-
induced and convection-induced water-side velocity scales, and the equation for $k_\mathrm{HE}$ ($\mathrm{m\,s^{-1}}$) is

$$k_\mathrm{HE} = ((C_1 U_{1.5})^2 + (C_2 w_*)^2)^{0.5} Sc^{-0.5}, \tag{13}$$

where $U_{1.5}$ is wind speed at 1.5 m height, $w_* = (-\beta z_\mathrm{AML})^{1/3}$ is the penetrative convection velocity, and $C_1 = 1.5 \times 10^{-4}$ and
$C_2 = 0.07$ are experimental dimensionless constants.

The fourth model applied in our study was a wind-based regression by MacIntyre et al. (2010), in which the gas transfer
velocity $k_\mathrm{MI}$ ($\mathrm{cm\,h^{-1}}$) is calculated separately for heating and cooling conditions as

$$k_\mathrm{MI} = \begin{cases} (2.04U_{10} + 2.0)\left(\frac{Sc}{600}\right)^{-0.5} & \text{if } \beta < 0, \\ (1.74U_{10} - 0.15)\left(\frac{Sc}{600}\right)^{-0.5} & \text{if } \beta \geq 0. \end{cases} \tag{14}$$

## 2.2 Model application

We used the MyLake C application to Lake Kuivajärvi presented in Kiuru et al. (2018) as the basis of the study. The model
setup, including model forcing data and the initial in-lake conditions, is nearly identical to that described in Kiuru et al. (2018).
The minor differences are pointed out in Sect. 2.2.2.



### 2.2.1 Study lake

Lake Kuivajärvi is an oblong, mesotrophic, and humic lake located in southern Finland (61° 50' N, 24° 16' E) at the vicinity of the SMEAR II station (Station for Measuring Ecosystem–Atmosphere Relations; Hari and Kulmala (2005)). The length of the lake is 2.6 km, the maximum width is 0.3 km, and the surface area is 0.63 km$^2$. The north-south-oriented lake has two distinct basins. The maximum depth of the deeper southern basin is 13.2 m (Heiskanen et al., 2014), which is more than double the mean depth 6.3 m. A measurement platform (Lake-SMEAR) is situated close to the deepest region of the lake. The approximate retention time of the lake is 0.65 years. Lake Kuivajärvi is surrounded by managed mixed coniferous forest together with small open wetland areas (Miettinen et al., 2015). The majority of the catchment area (9.4 km$^2$) of the lake is flat. The main inlet stream with a mean pH of 6.5 (Dinsmore et al., 2013) drains four upstream lakes, which are smaller in area than Lake Kuivajärvi. The lake is dimictic: the spring turnover usually occurs rapidly right after ice-off in late April or early May, and the summer stratification period lasts until the autumn turnover in September or October. The duration of the ice-covered period and the concomitant inverse stratification is usually 5–6 months (Heiskanen et al., 2015). The turnover periods are hot moments for the release of $CO_2$ accumulated in the hypolimnion of the lake during stratification (Miettinen et al., 2015). Because of high terrestrial inputs of organic matter, a median concentration of DOC in the surface water is 12–14 mg L$^{-1}$ (Miettinen et al., 2015) and water clarity is rather low, a median light attenuation coefficient $K_L$ being around 0.6 m$^{-1}$ (Heiskanen et al., 2015).

### 2.2.2 Model forcing and calibration data

The meteorological forcing data and hydrological loading data used in the model application are described in detail in Kiuru et al. (2018). The daily averages of wind speed at 1.5 m and incoming shortwave radiation together with in-lake temperature and $CO_2$ concentration were obtained from automatic platform measurements (Heiskanen et al., 2014; Mammarella et al., 2015), and the remaining meteorological forcing data were obtained from the SMEAR II station or from weather stations (Finnish Meteorological Institute) in Hyytiälä located less than 1 km from the lake (precipitation) and in Tikkakoski located approximately 95 km to the north-east from the lake (cloud cover fraction). Differing from Kiuru et al. (2018), the $CO_2$ mixing ratio in the atmosphere was assumed to be 395 ppm on the basis of the rather fragmentary time series of the high-frequency in situ measurements of the $CO_2$ mixing ratio, the method of which is described in Erkkilä et al. (2018).

The construction of the time series for lake inflow was based on continuous measurements of the discharges at the main inlet and at the outlet of Lake Kuivajärvi in 2013–2014 (Dinsmore et al., 2013). Because the total measured outflow volumes were approximately double the main inlet discharge volumes on an annual scale, the daily inflow volumes were corrected by a factor of 2 in order to include the potential contributions of smaller inlet streams and groundwater to lake inflow. At the main inlet, water temperature was measured approximately two times a month in 2013 and continuously in 2014 and $CO_2$ concentration was measured two times a month in 2013 but mostly at intervals of 2–3 d around the period of ice-off in April and May using the procedure described in Miettinen et al. (2015). Daily time series were generated by linear interpolation.





The model was calibrated against the daily averages of the automatic high-frequency $CO_2$ concentration measurements. In addition, the automatic water column temperature measurements were used in model performance validation. The $CO_2$ concentrations were measured at 0.2, 1.5, 2.5, and 7.0 m, and the temperature measurements were performed at 0.2 m, at 0.5 m intervals from 0.5 to 5.0 m, and at 6, 7, 8, 10 and 12 m using the measurement systems described in Heiskanen et al. (2014)
and Mammarella et al. (2015).

### 2.2.3 Model assessment data

We used additional meteorological measurements in assessing the performance of the alternative models for $k$ incorporated into MyLake C during the period May–October 2013. An EC system located on the measurement platform measures the turbulent fluxes of momentum, heat, and water vapor ($H_2O$) over the lake (Mammarella et al., 2015). The EC flux measurement system
includes an ultrasonic anemometer (USA-1, Metek GmbH, Elmshorn, Germany) and a closed-path infrared gas analyzer (LI-7200, LICOR Inc., Nebraska, USA) for measuring $CO_2$ and $H_2O$ mixing ratios at 1.8 m height above the lake surface. Air temperature and relative humidity were measured with a Rotronic MP102H/HC2-S3 (Rotronic Instrument Corp., NY) and radiation components with a CNR1 net radiometer (Kipp & Zonen, Delft, Netherlands). Automatic platform measurements of net surface longwave radiation and EC measurements of sensible heat flux, $H_2O$ flux, and momentum flux were used in the
determination of net surface heat flux and atmospheric friction velocity. During EC data post-processing, latent heat flux was calculated from the $H_2O$ flux, and the atmospheric friction velocity was derived from the momentum flux. All EC measurement data were given as half-hour block averages. The EC measurements are explained in more detail in Erkkilä et al. (2018), and the description of EC data post-processing is found in Mammarella et al. (2015) and Mammarella et al. (2016). Contrary to the model forcing data, the air temperatures that were used in the measurement-based determination of the gas transfer velocities
were obtained from the platform measurements instead of the SMEAR II station when platform measurements were available. In addition, the rather intermittent platform measurement data on relative humidity were used. Missing relative humidities were replaced by a value of 75 % in the calculation of the water-side friction velocity.

There was a gap in the heat flux data on 14–27 June because of EC system malfunction, and some of the existing data were discarded through the application of EC quality screening criteria presented in Erkkilä et al. (2018). In addition, only wind di-
rections along the lake (130°–180° and 320°–350°) were included so that heat fluxes from the surrounding land were excluded. The monthly data coverage was 43–69 % and 32–70 % of the original data for sensible and latent heat fluxes, respectively. We constructed gap-filled half-hour time series for sensible and latent heat fluxes using linear fits between the measured sensible heat flux and wind speed multiplied by the air–surface water temperature difference and between the measured latent heat flux and wind speed multiplied by the vapor pressure difference, according to Mammarella et al. (2015). Only the vapor pressures
calculated from the measured relative humidities were used in the latter fit. The fitting was performed independently for each month.

We compared the simulated gas transfer velocities for $CO_2$ and the simulated air–water $CO_2$ fluxes to those determined directly from measurements using the corresponding gas exchange models. The latter are hereinafter referred as to calculated gas transfer velocities and calculated $CO_2$ fluxes. The calculated $CO_2$ transfer velocities for each of the four gas exchange



models were obtained using the daily averages of required measured variables. The calculated air–water $CO_2$ fluxes were further obtained as the product of the calculated $CO_2$ transfer velocities and the daily averages of the measured air–water $CO_2$ concentration gradient. The conditions were thus compatible with the daily time step applied in MyLake C. The atmospheric equilibrium concentrations of $CO_2$ were calculated from the measured atmospheric $CO_2$ mixing ratios. The daily averages

of the depth of the AML were estimated from the daily averaged temperature profiles as the depth at which water column temperature was within $0.25\,^{\circ}\mathrm{C}$ of the temperature at $0.2\,\mathrm{m}$ as in Erkkilä et al. (2018). Following Mammarella et al. (2015), a value of $2\,\mathrm{m}^{-1}$ was used for the total attenuation coefficient of shortwave radiation $K_{\mathrm{L}}$ in the calculation of $Q_{\mathrm{eff}}$.

### 2.2.4   Model calibration and validation

We estimated the MyLake C parameters utilizing a Markov chain Monte Carlo-based Bayesian inference algorithm following

the procedures in the original calibration of the Lake Kuivajärvi application presented in Kiuru et al. (2018). Each of the four new versions of the MyLake C Lake Kuivajärvi application, using the models for $k$ by Cole and Caraco (1998) (both the MyLake C version and the respective gas exchange model being hereinafter referred to as CC), Heiskanen et al. (2014) (HE), MacIntyre et al. (2010) (MI), and Tedford et al. (2014) (TE), was calibrated individually. The simulations with the MyLake C versions using different gas exchange models are hereinafter collectively referred to as GEMs. The model grid length was

$0.5\,\mathrm{m}$. The model was run from 8 January 2013 to 31 December 2014. The calibration period extended from 8 January to 31 December 2013, and the measurements in 2014 were used for model validation.

The calibrations were performed against the daily averages of the automatic water column $CO_2$ concentration measurements at the depths of 0.2, 2.5, and $7\,\mathrm{m}$. We chose to apply the automatic measurements instead of the corresponding manual measurements used in the model calibration in Kiuru et al. (2018) because the calculation of daily $CO_2$ fluxes was based on

the automatic measurements at $0.2\,\mathrm{m}$ in this study and the simulation results were thus comparable with the calculated $CO_2$ fluxes. Even though the near-surface $CO_2$ concentration was the most significant factor considering air–water $CO_2$ exchange, deeper depths were included so that model behavior would remain reasonable also at deeper levels.

The calibrated model parameters were selected on the basis of the original calibration. However, because the new calibrations were not performed against water column DO concentrations, the parameters related to interactions between DO and

$CO_2$ were excluded from the parameter set. The DIC inflow concentration scaling factor $C_{\mathrm{DI,IN}}$, applied during open water seasons, was introduced as a new calibration parameter. The other parameters included in the calibration were the vertical turbulent diffusion parameter $a_{\mathrm{k}}$, the wind sheltering coefficient $W_{\mathrm{str}}$, the DOC-related specific attenuation coefficient of photosynthetically active radiation $\beta_{\mathrm{DOC}}$, the maximal phytoplankton growth rate at $20\,^{\circ}\mathrm{C}$ $\mu'_{20}$, the phytoplankton death rate at 20 $^{\circ}\mathrm{C}$ $m_{20}$, the degradation rates of labile DOC $k_{\mathrm{DOC,1}}$ and semilabile DOC $k_{\mathrm{DOC,2}}$, the fragmentation rates of autochthonous

POC $k_{\mathrm{POC,1}}$ and allochthonous POC $k_{\mathrm{POC,2}}$, and the sedimentary POC degradation rate $k_{\mathrm{POC,sed}}$. The parameters obtained in the original calibration, or the default parameters, were used as the means of the prior parameter distributions.

One parameter chain with 3000 iterations was produced in each calibration. The starting points were set to 50th percentiles of the prior distributions. The first half of each resultant chain was discarded as a burn-in period, and the final parameters chains included 1500 parameter sets. The medians of the final posterior distributions (Figs. S1–S4) were chosen as the calibrated





parameters. They are presented, together with the default parameters, in Table 1. After the calibrations, additional goodness-of-fit metrics were calculated. The Nash–Sutcliffe efficiency (NS) gives a relative evaluation assessment, determining the relative magnitude of the residual variance compared to the variance of measurement data (Moriasi et al., 2007). The value of the normalized bias ($B^*$) describes a systematic overestimation ($B^* > 0$) or underestimation ($B^* < 0$) of a state variable in the

simulation, whereas the normalized unbiased root-mean-square difference ($\mathrm{RMSD}'^*$) shows if the standard deviation of the simulated values is higher ($\mathrm{RMSD}'^* > 0$) or smaller ($\mathrm{RMSD}'^* < 0$) than that of the measurements (Los and Blaas, 2010).

### 2.2.5  Calculation of $CO_2$ budgets

After the calibrations, we calculated $CO_2$ budgets for the epilimnion of the lake during the periods of continuous summer stratification in 2013 and 2014 for each GEM. The epilimnion was defined as the layer in which water temperature was within

$1\,^{\circ}\mathrm{C}$ of surface temperature. The stratified period was defined to begin on the day of the formation of the thermocline after ice-off and to finish when the depth of the epilimnion ($z_{\mathrm{epi}}$) reached the value of $7\,\mathrm{m}$ in the simulations. The exchange of $CO_2$ between the epilimnion and the atmosphere is balanced in MyLake C by (1) net external loading of $CO_2$, (2) net epilimnetic $CO_2$ production, and (3) the release of $CO_2$ from deeper layers to the epilimnion. The net external loading equals the amount of terrestrially produced $CO_2$ entering the lake via stream inflow subtracted by the amount of $CO_2$ in lake outflow. The release

of $CO_2$ from the metalimnion or the hypolimnion occurs through deepening of the epilimnion due to wind-induced mixing or thermal convection. If the epilimnetic volume becomes smaller, a portion of $CO_2$ is again confined below the epilimnion and the amount of $CO_2$ in the remaining epilimnion is reduced.

## 3  Results

### 3.1  Model calibration

Even though the differences between the formulations of the gas exchange models incorporated into MyLake C are rather notable, the resultant $CO_2$ concentrations did not differ substantially between the GEMs (Fig. 1). However, an optimal simulation result can be attained through many different combinations of processes related to in-lake carbon dynamics and fluvial and atmospheric exchange in MyLake C, which is seen in the variation between the parameter values obtained from the different calibrations (Table 1). The calibrations were performed only against $CO_2$ concentrations, and the aim of the calibration

was not to try to reproduce the actual in-lake carbon cycling but rather to compare different possible ways to generate an optimal water column $CO_2$ concentration. The performance metrics for $CO_2$ concentration shown in the Supplement (Table S1) indicate that all GEMs yielded too low $CO_2$ concentrations ($B^* < 0$) at all depths during the calibration and validation periods with only few exceptions. However, the $CO_2$ concentration measurements performed during the ice-covered periods were largely not applicable at $0.2\,\mathrm{m}$ because of the lake ice cover and sometimes inapplicable also at deeper levels because of

system malfunction.





The average near-surface (0–0.5 m) $CO_2$ concentrations over the open water seasons were notably higher in CC (44.3 mmol m$^{-3}$ and 40.3 mmol m$^{-3}$ in the calibration year 2013 and in the validation year 2014, respectively) than in the other GEMs (HE: 34.2 mmol m$^{-3}$ and 31.6 mmol m$^{-3}$; MI: 31.5 mmol m$^{-3}$ and 29.4 mmol m$^{-3}$; TE: 36.9 mmol m$^{-3}$ and 34.1 mmol m$^{-3}$). Only the days with applicable corresponding water column $CO_2$ concentration measurement data were included

in the averaging. By contrast, the open water season averages of the measured near-surface (0.2 m) $CO_2$ concentrations were 45.2 mmol m$^{-3}$ in 2013 and 37.2 mmol m$^{-3}$ in 2014. Thus, CC yielded a higher near-surface $CO_2$ concentration compared to the measurements in 2014 when only the ice-free season, the period of air–water $CO_2$ exchange, is considered. The simulated open water seasons were determined from the simulated ice-off and ice-on dates. Because $CO_2$ flux differs from zero starting from the day after ice-off in MyLake C, the simulated open water seasons applied in the study were 3 May–25 November 2013

and 16 April–22 November 2014. In 2013, the observed open water season lasted from 1 May to 27 November. In 2014, the observed ice-off date was 12 April.

The simulated $CO_2$ transfer velocities and air–water $CO_2$ fluxes are presented in Fig. S5. The yearly average values of $k$ were lowest in CC and rather similar between the other GEMs (CC: 2.81 cm s$^{-1}$ and 2.76 cm s$^{-1}$ for the calibration period and the validation period, respectively; HE: 5.44 cm s$^{-1}$ and 5.33 cm s$^{-1}$; MI: 5.87 cm s$^{-1}$ and 5.82 cm s$^{-1}$; TE: 4.73 cm s$^{-1}$ and

4.66 cm s$^{-1}$). The differences in the simulated fluxes between GEMs were dissimilar to those in $k$ because of the differences in the simulated near-surface $CO_2$ concentrations. The smallest $k$ values in CC were compensated by the highest near-surface $CO_2$ concentrations. By contrast, a high daily $CO_2$ efflux due to a high $k$ in MI reduced the simulated near-surface $CO_2$ concentration compared to the other GEMs during the whole simulation period. Overall, the differences in yearly air–water $CO_2$ fluxes between GEMs were smaller than those in the values of $k$ (CC: 0.22 µmol m$^{-2}$ s$^{-1}$ and 0.20 µmol m$^{-2}$ s$^{-1}$

for the calibration period and the validation period, respectively; HE: 0.28 µmol m$^{-2}$ s$^{-1}$ and 0.26 µmol m$^{-2}$ s$^{-1}$; MI: 0.25 µmol m$^{-2}$ s$^{-1}$ and 0.24 µmol m$^{-2}$ s$^{-1}$; TE: 0.28 µmol m$^{-2}$ s$^{-1}$ and 0.27 µmol m$^{-2}$ s$^{-1}$).

The $CO_2$ efflux during the first few days after ice-off was higher in GEMs with a high $k$, which increased the water column pH in comparison to CC. The differences remained rather constant during most of the open water seasons. The near-surface pH was on average 0.20–0.26 and 0.18–0.25 units higher in the other GEMs than in CC during the open water seasons of 2013 and

2014, respectively. As a result, the average fractions of $CO_2$ of DIC in the near-surface layer were, respectively, 6–8 and 5–6 percentage units higher in CC than in other GEMs, which also contributed to the higher near-surface $CO_2$ concentration in CC than in other GEMs. In addition, the open water season average near-surface pH was 0.22 units higher in 2014 than 2013 in all GEMs. Accumulation of bicarbonate in the water column in the course of the simulations may have resulted in an excessively high pH and thus a relatively lower $CO_2$ concentration in 2014 compared to 2013.

The differences in simulated temperatures between GEMs, primarily due to different attenuation of shortwave radiation in the water column, were rather small especially at 0.2 m and at 2.5 m (Fig. S6). High epilimnetic concentrations of both Chl $a$ and DOC, resulting from a low phytoplankton death rate and a high allochthonous POC fragmentation rate, respectively, in MI resulted in the strongest attenuation of shortwave radiation and thus the highest near-surface temperature because of a thinner and warmer epilimnion than in other GEMs. The open water season average near-surface temperatures were 0.28–0.47

°C and 0.65–0.86 °C lower than the corresponding measured averages in the calibration and validation periods, respectively,





being highest in MI and lowest in TE. The differences were greatest in November before ice-on. The simulated near-surface temperatures tended to be somewhat too low in spring and early summer during both periods and somewhat too high in the late summer and autumn of the calibration year.

Heat transfer to the depth of 7 m right after the onset of the summer stratified period was insufficient in the calibration year in all GEMs, and small values of $a_k$ also reduced heat transfer through the epilimnion during summer stratification. As a result, water column temperature remained too low at the depth of 7 m, which was located in the hypolimnion for most of the summer, during the stratified period in the simulations. However, the performance of the simulation of $CO_2$ concentration was successful also at the depth of 7 m. The summertime mixed layer thickness was rather similar between GEMs during the calibration year but more variable during the validation year. Simulated thermocline deepening matched the measurements during the late summer of the calibration year but was too early in the validation year. The deepening was slowest in HE because a somewhat stronger temperature gradient in the metalimnion, which was due to the smallest $a_k$, resisted wind-induced thermocline erosion during summer.

## 3.2 Effective heat flux

The effective heat fluxes at the air–water interface simulated with each GEM on 3 May to 31 October 2013 and the corresponding values calculated on the basis of heat flux and radiation measurements are presented in Fig. 2a. The largest differences between the magnitudes and the directions of simulated and measured $Q_{eff}$ were seen in early May. The simulated $Q_{eff}$ was directed out of the lake throughout the study period except for few occasions in early May and in October, whereas measurement-based calculations yielded more frequent occurrences of a positive daily $Q_{eff}$. Also, a negative $Q_{eff}$ was often overestimated by the simulations because of overly high negative sensible and latent heat fluxes and net longwave radiation (Fig. S7). The performance of the simulation of the components of surface heat flux was rather poor (Table S2). Overall, the $Q_{eff}$ simulation performance was not very good ($R^2 = 0.39$–$0.41$, RMSE = $48.2$–$49.2$ W m$^{-2}$, NS = $0.11$–$0.14$, $B^* = -0.47 \ldots -0.46$, $n = 164$). The differences in the simulated $Q_{eff}$'s between GEMs, resulting mainly from different surface temperatures, were quite small.

The extent of shortwave radiative heating of the AML, $Q_{SW,AML}$, is dependent on $z_{AML}$. The simulated $z_{AML}$ was greater than the measured daily average with few exceptions at the beginning and near the end of the study period (Fig. 2b), which increased the simulated $Q_{SW,AML}$ and decreased a negative $Q_{eff}$. The simulation with a daily time step generated clear temperature variation in the epilimnion only on days with a high amount of surface heating in early summer and midsummer, which resulted in an overly deep AML during most of the period. In addition, the model with a sequential description of thermal processes did not catch simultaneous wind mixing and surface heat exchange processes that resulted in a deeper observational AML in spring and late autumn. However, day-to-day variation in the discrepancy of $Q_{SW,AML}$ was high throughout the study period. Also, the simulations highly underestimated the atmospheric friction velocity ($R^2 = 0.35$, RMSE = $0.11$ m s$^{-1}$, NS = $-3.2$, $B^* = -1.89$, $n = 166$) (Fig. S8), the simulated $u_{*a}$ being on average only 46 % of the measured daily average. The simulated daily drag coefficient $C_d$ at 1.5 m was affected by atmospheric stability conditions. The median $C_d$ varied from $1.589 \times 10^{-3}$ to $1.593 \times 10^{-3}$ between the GEMs.




### 3.3 CO$_2$ exchange

The differences between simulated gas transfer velocities for $CO_2$ and the respective calculated values during the study period 3 May–31 October 2013 were rather small in the cases of gas exchange models based solely on wind speed, CC and MI, but the discrepancies were higher in HE and TE, which include also the effect of thermal convection on gas exchange (Fig. 3, Table S3). The simulations with CC and MI often yielded slightly higher values of $k$ than the respective calculations because the simulated surface temperature was higher than the measured daily average (Fig. S6) and thus the temperature-dependent Schmidt number correction of $k$ was different. Also, the occurrences of a simulated negative $\beta$ in early May in MI yielded higher $k_{MI}$'s compared to the respective calculated values obtained from the observed positive $\beta$. The simulated $k_{HE}$ was often higher than the calculated counterpart because of a high negative $Q_{eff}$ or a deep AML in the simulations (Fig. 2), which resulted in a high penetrative convection velocity. In HE, the effects of wind-induced shear and thermal convection on $k$ are set to be roughly of the same order of magnitude and the wind-induced shear velocity is calculated from wind speed, whereas $CO_2$ flux is driven principally by wind shear, which is calculated directly from $u_{*a}$, in TE. Because the simulated $u_{*a}$ was consistently significantly lower than the corresponding daily measured average, the simulated $k_{TE}$ was on average 40 % lower than the calculated value.

The simulated near-surface $CO_2$ concentrations were significantly too low during most of the study period in all GEMs except for CC, which yielded too high concentrations in autumn (Fig. 4, Table S3). The higher the simulated $k$ and daily $CO_2$ efflux, the greater was the resulting decrease in near-surface $CO_2$ concentration. The decline of near-surface $CO_2$ concentration after ice-off was too rapid in all GEMs, especially in MI, the GEM with the highest $k$. Simulated near-surface $CO_2$ concentration declined close to the atmospheric equilibrium concentration in all GEMs also in late summer because of insufficient gain of $CO_2$ in a shallow epilimnion developed under warm and calm conditions. The low simulated air–water $CO_2$ concentration gradients in May resulted in an underestimated air–water $CO_2$ flux from the water column compared to the respective calculated fluxes (Fig. 5, Table S3). The simulated flux was notably lower than the calculated flux in TE during the whole study period because of a small $k_{TE}$. On the contrary, CC notably overestimated the corresponding calculated $CO_2$ flux in August and September because of a high simulated near-surface $CO_2$ concentration. Also, the simulated $CO_2$ flux was slightly higher than the calculated flux in HE in August and September because of high epilimnetic net $CO_2$ production. The total simulated $CO_2$ flux during May–October matched the calculated flux in CC but was notably lower in HE and MI and less than half of the calculated flux in TE (Table 2). The underestimated near-surface $CO_2$ concentrations were somewhat compensated for by the higher simulated $k_{HE}$ and $k_{MI}$ compared to the calculated counterparts, which decreased the difference between the simulated and calculated fluxes in HE and MI.

The applied gas exchange models yielded notably different calculated monthly $CO_2$ effluxes (Table 2). The $CO_2$ fluxes were calculated using the measured air–water $CO_2$ concentration gradients, and thus the differences between the calculated fluxes were only due to different values of $k$. Monthly fluxes calculated with MI were nearly or even more than double of those calculated with the other wind-based model CC. Days with a positive $\beta$, resulting in a lower $k_{MI}$, occurred mainly in May and October, and thus the difference between the $CO_2$ fluxes calculated with MI and CC was slightly smaller in those months. The





models that include the effect of thermal convection, HE and TE, yielded notably higher $CO_2$ fluxes than the simplest model, CC. Nevertheless, the $CO_2$ fluxes calculated with MI were slightly higher than those calculated with HE. The $CO_2$ fluxes calculated with TE were clearly the highest in all months, which was, however, not the case in the simulations.

The calculated daily values of $k$ and $CO_2$ flux were dependent on the calculation interval. If the daily $k$ had been calculated as the daily average of calculated half-hour values of $k$ instead of using the daily averages of the input variables, the results would have been different. The daily averages of calculated half-hour $k_{MI}$ (RMSE = 0.70 cm h$^{-1}$, $B^* = -0.16$) and $k_{TE}$ (RMSE = 0.22 cm h$^{-1}$, $B^* = -0.04$) were lower than the respective values calculated using daily averages of input variables, whereas the opposite was the case for $k_{HE}$ (RMSE = 0.48 cm h$^{-1}$, $B^* = 0.20$) and $k_{CC}$ (RMSE = 0.16 cm h$^{-1}$, $B^* = 0.15$). On the contrary, the calculation of a daily $CO_2$ flux as the average of half-hour fluxes yielded a slightly higher $CO_2$ flux in all GEMs (HE: RMSE = 0.066 µmol m$^{-2}$ s$^{-1}$, $B^* = 0.13$; CC: RMSE = 0.034 µmol m$^{-2}$ s$^{-1}$, $B^* = 0.11$; MI: RMSE = 0.10 µmol m$^{-2}$ s$^{-1}$, $B^* = 3.4 \times 10^{-4}$; TE: RMSE = 0.11 µmol m$^{-2}$ s$^{-1}$, $B^* = 0.05$).

The differences resulting from the different methods of the calculation of a daily $k$ can partly be explained by the behavior of the driving variables of the models. Using the daily averages of the input variables in the calculation may have smoothened out the effects of the spells of stronger negative buoyancy flux or a deeper AML that increase the half-hour $k_{HE}$ and the effects of the occasions of positive buoyancy flux that decrease the half-hour $k_{MI}$. Daily averaging of wind speed may have cut out the rapid increase of $k_{CC}$ under stronger-wind conditions during the course of day due to the greater-than-linear dependence of $k_{CC}$ on wind speed. By contrast, because the dependence of $k_{TE}$ on $u_{*a}$ is less than linear and the impact of thermal convection on $k_{TE}$ is minor, the effect of the diel variation of $u_{*a}$ and thus the relative difference between the methods of the calculation of $k_{TE}$ was rather small.

## 3.4 Lake $CO_2$ budgets

The simulated $CO_2$ budgets for the epilimnion of the lake during the periods of continuous summer stratification in 2013 and 2014 differed between GEMs as a response to different $CO_2$ effluxes (Table 3). The simulations were not able to reproduce the short-lived episodes of a very shallow epilimnion on days with high solar radiation and low wind speeds in late August and early September 2013, but at other times the simulated $z_{epi}$ matched rather well the depths estimated from the measured daily temperature profiles (Fig. 6). The epilimnion formed 11 d earlier and extended to 7 m 16–22 d later in 2013 than in 2014. The in-lake $CO_2$ concentrations were higher at the onset of stratification in 2013 than in 2014 because of less effective water column ventilation during the shorter spring mixing period. As a result, the amount of $CO_2$ in the epilimnion decreased during the stratified period in 2013, whereas it increased slightly in 2014.

A higher net in-lake $CO_2$ production or a higher terrestrial $CO_2$ load was required to compensate for the higher $CO_2$ efflux in GEMs that yielded higher values of $k$ (Table 3). Phytoplankton concentration, regulated in MyLake C by the growth and death rates $\mu'$ and $m$, respectively, impacts $CO_2$ dynamics both directly through the amount of carbon fixation and indirectly through changes in epilimnetic thermal structure due to attenuation of solar radiation. A high $\mu'$ resulted in faster phytoplankton growth in spring and thus in an earlier occurrence of the spring bloom, and a small $m$ resulted in a higher phytoplankton biomass during midsummer and late summer. HE and MI yielded the highest maximum near-surface Chl $a$ concentrations, approximately 15





mg m$^{-3}$ in 2013 and close to 20 mg m$^{-3}$ in 2014. In CC and TE, Chl $a$ concentrations were greater than 10 mg m$^{-3}$ during the growth peaks but less than 5 mg m$^{-3}$ at other times because of the high values of $m$. The open water season average near-surface Chl $a$ concentration was highest in HE (9.6 and 9.3 mg m$^{-3}$ in 2013 and 2014, respectively), followed by MI (7.5 and 6.1 mg m$^{-3}$), CC (3.9 and 4.0 mg m$^{-3}$), and TE (2.3 and 2.1 mg m$^{-3}$).

However, a high phytoplankton biomass did not imply high $CO_2$ consumption because of phosphorus limitation of phytoplankton growth in the model and the resultant reduction of photosynthetic $CO_2$ consumption under high Chl $a$ and low bioavailable phosphorus concentrations in the simulations. Instead, $CO_2$ fixation occurred at a steady rate and the total $CO_2$ consumption over the whole growing season was relatively higher under a low Chl $a$ concentration due to a high $m$. The highest average phytoplankton biomass in HE resulted in the highest $CO_2$ fixation; however, also total net $CO_2$ production was

highest in HE because of high $k_{\mathrm{POC},1}$ and $k_{\mathrm{DOC},1}$. Small values of $k_{\mathrm{POC},1}$ and $k_{\mathrm{DOC},2}$ resulted in a relatively low net $CO_2$ production despite low $CO_2$ fixation and a high $k_{\mathrm{POC,sed}}$ in TE. Net $CO_2$ production was lowest in CC because of rather high total $CO_2$ fixation during the long growing season and the rather small $k_{\mathrm{POC},1}$ and $k_{\mathrm{DOC},1}$.

A considerable increase in the inflow DIC concentration by way of the scaling factor was essential in order to significantly increase the terrestrial $CO_2$ input to the lake in GEMs with a high $CO_2$ efflux. The measured inflow $CO_2$ concentration was

200–250 mmol m$^{-3}$ until ice-off, less than 80 mmol m$^{-3}$ during May, and mainly between 50 and 100 mmol m$^{-3}$ during the summer and autumn. Thus, the default inflow $CO_2$ concentration was only approximately double the simulated near-surface $CO_2$ concentrations during most of the open water season, and the effect of external $CO_2$ loading on in-lake $CO_2$ concentration was inevitably rather small especially during the low-discharge period in late summer and autumn. The values of $C_{\mathrm{DI,IN}}$ determined the order of the amounts of the net external $CO_2$ load to the lake in the GEMs (TE: 42000 and 45000

kg $CO_2$ over the years 2013 and 2014, respectively; MI: 27500 and 31400 kg $CO_2$; HE: 26500 and 30600 kg $CO_2$; CC: 22200 and 25800 kg $CO_2$).

However, the total net external $CO_2$ loads to the lake over the stratification periods were slightly higher than the net external $CO_2$ loads to the epilimnion in Table 3 because stream inflow was directed into the metalimnion on days when the inflow temperature was lower than the epilimnetic temperature. The epilimnetic loads were 90–92 % and 98–99 % of the total loads

in 2013 and 2014, respectively, the proportions being highest in CC and lowest in MI. The amount of $CO_2$ outflow was relatively large in CC because of the high epilimnetic $CO_2$ concentration; thus, the net external $CO_2$ load was relatively lower in CC than in other GEMs compared to the differences in $C_{\mathrm{DI,IN}}$. In addition, because inflow pH was unaltered in the scaling of inflow DIC concentration, some of the increased $CO_2$ load was eventually evaded to the atmosphere in the simulations but the bicarbonate fraction of DIC remained in the water column, which resulted in a slight increase in in-lake pH and a decline

in the $CO_2$ fraction of DIC especially in GEMs with a high $k$. Nevertheless, the impact of different amounts of bicarbonate loading on the in-lake pH was minor compared to the impact of different springtime $CO_2$ effluxes between GEMs.



## 4   Discussion

### 4.1   Differences between calculated and simulated $CO_2$ fluxes

There was less variation between the air–water $CO_2$ fluxes simulated with different GEMs than between the $CO_2$ fluxes calculated with the corresponding different gas exchange models on the basis on measured surface heat fluxes and air–water

$CO_2$ concentration gradients (Table 2). This was caused both by differences between the simulated and calculated values of $k$ and by insufficient epilimnetic $CO_2$ production in the simulations. An increased terrestrial $CO_2$ loading or an increased in-lake $CO_2$ production was needed to balance the higher $CO_2$ loss from the epilimnion through efflux in GEMs with a higher $k$ compared to the simple wind-based CC (Table 3). Still, the simulations yielded too low near-surface $CO_2$ concentrations (Fig. 4, Table S3), which contributed to the underestimation of $CO_2$ fluxes (Fig. 5). Calibrating the model only against the

near-surface $CO_2$ concentration and thus using even higher values for organic carbon fractionation and degradation parameters would have improved the performance of the simulation of epilimnetic $CO_2$ concentration; however, it would have resulted in uncontrollable and probably excessively high $CO_2$ concentrations in deeper layers, which is disadvantageous in a year-round, vertically layered lake model.

The day-to-day performance of the simulation of epilimnetic $CO_2$ concentration was also partly determined by the simulated

thermal stratification and epilimnetic volume. The simulations generally yielded too low a near-surface $CO_2$ concentration when the simulated $z_{\text{epi}}$ was in accordance with the observed depth and performed more adequately only during periods when the simulated $z_{\text{epi}}$ was too high (Figs. 4 and 6). The measurements showed an increase in the near-surface $CO_2$ concentration when the epilimnion became thicker, and vice versa, during the stratified period in 2013. Thermocline tilting-induced upwelling and convection-induced entrainment transported more $CO_2$-rich water into the epilimnion on windy and cool days (Heiskanen

et al., 2014). Conversely, high solar radiation input combined with calm conditions results in the warming of near-surface water and the formation of a thin epilimnion with a lower $CO_2$ concentration. High solar radiation also enhances photosynthesis and thus increases the uptake of $CO_2$ (Provenzale et al., 2018). An overly deep simulated epilimnion resulted in enhanced $CO_2$ release from deeper layers and a higher total net $CO_2$ production in a larger epilimnetic volume, which were able to compensate for the $CO_2$ efflux in the simulations.

The accuracy of the determination of a daily $Q_{\text{eff}}$ and the applicability of the concept of a daily AML are issues that may cause uncertainties when the gas exchange models are used either to calculate or to simulate daily estimates of $k$. The calculated half-hour $Q_{\text{eff}}$ was generally directed into the lake on some occasions at daytime because of solar heating of the AML and always directed out of the lake at nighttime, and $z_{\text{AML}}$ often increased during nighttime and decreased under radiative heating of near-surface water at daytime. Boundary layer models and surface renewal models have been developed to describe short-

term dynamics of turbulence in a shallow AML, and thus they may not perform equally well in calculations with a daily time step.

The wind-based CC yielded the lowest and the surface renewal model TE the highest calculated air–water $CO_2$ fluxes, which is in line with the comparisons of different gas exchange models using data from Lake Kuivajärvi by Mammarella et al. (2015) and Erkkilä et al. (2018); however, the differences in simulated $CO_2$ fluxes between CC and other GEMs were notably





smaller than the corresponding differences in the two experimental studies. The performance of TE is strongly dependent on the magnitude of $u_{*a}$ because wind shear is highly dominant over thermal convection as the generator of turbulence in the model. Because the simulations yielded significantly lower $u_{*a}$ compared to the values obtained through EC measurements (Fig. S8), the $CO_2$ flux obtained with TE was much lower than the corresponding calculated flux. Also Erkkilä et al. (2018) found that

$u_{*a}$ calculated from wind speed was lower than the measured $u_{*a}$ in Lake Kuivajärvi. Bulk models for surface stress may yield low values for $u_{*a}$ over a lake especially when parameterized for open sea conditions with low surface roughness (Wang et al., 2015), which is the case in MyLake C. Lake size may also affect the relative differences between gas transfer velocities obtained with different gas exchange models. Dugan et al. (2016) applied different gas exchange models to the calculation of DO exchange in temperate lakes of various sizes. Simple, wind-based models yielded clearly lower values of $k$ than more complex

models in lakes similar to Lake Kuivajärvi in size, whereas the differences between the model types were smaller in larger lakes with generally higher wind speeds and a higher relative importance of wind-induced mixing compared to convection. In addition, ecosystem-specific empirical regression models may not be suitable for lakes with dissimilar characteristics (Vachon and Prairie, 2013).

## 4.2  Comparison to EC $CO_2$ flux measurements

Estimates of air–water $CO_2$ fluxes obtained with the gas exchange models applied in our study have been compared with 30-min block-averaged EC $CO_2$ flux measurements over Lake Kuivajärvi (Heiskanen et al., 2014; Mammarella et al., 2015; Erkkilä et al., 2018). Heiskanen et al. (2014) compared the half-hour $k$'s calculated with HE, CC, and MI with those obtained through EC measurements of $CO_2$ flux in August–November 2011. In the study, the average values of $k_{HE}$ and $k_{MI}$ were approximately 70 % of the corresponding measurement-based values, but the average $k_{CC}$ was only about half of the average

$k_{HE}$ and $k_{MI}$. Erkkilä et al. (2018) compared the daily medians of EC $CO_2$ flux during a two-week period in October 2014 with the daily median $CO_2$ fluxes calculated with CC, HE, and TE. The $CO_2$ fluxes obtained with HE and TE were 60 % of the EC $CO_2$ fluxes and approximately double the $CO_2$ fluxes obtained with CC. Overall, TE yielded the best correspondence with the EC fluxes. TE outperformed CC also in the comparison of half-hour $CO_2$ fluxes during the open water periods of 2010 and 2011 in Mammarella et al. (2015). In our study, the best agreement with simulated and calculated $CO_2$ fluxes was found in CC,

whereas TE yielded the lowest simulated fluxes in comparison to the corresponding calculated fluxes. Thus, none of the GEM outputs can be considered compatible with EC $CO_2$ fluxes, provided that the conclusions from the half-hour comparisons in the above-mentioned studies can be extended to a daily scale.

The simulation results for the daily air–water $CO_2$ fluxes cannot be directly compared with EC data because the data coverage of EC flux measurements is often low. For example, the data coverages for $CO_2$ flux were 27 % and 37 % in Erkkilä

et al. (2018) and Mammarella et al. (2015), respectively. Quality screening excludes much of the measurement data, and short-time system malfunction may cause significant data loss during long study periods. Daily average or median EC $CO_2$ flux may not be representative for the whole day because of the temporal bias of the measurements. EC flux measurements often tend to be inapplicable especially at nighttime because of flux nonstationarity during light winds and cooling (Heiskanen et al., 2014) or advection of $CO_2$ from the surrounding forest (Erkkilä et al., 2018). EC $CO_2$ fluxes over boreal lakes are often enhanced



at night by water-side convection (Podgrajsek et al., 2015) or because of a higher air–water $CO_2$ concentration gradient due to the absence of photosynthesis as a $CO_2$ sink (Erkkilä et al., 2018).

Both the calculated and the simulated values of $k$ were determined by means of the platform data. They were thus suitable for comparison with each other but, however, may not represent the average conditions over the lake and hence may not yield

correct estimates of whole-lake $CO_2$ fluxes. Wind speed, $u_{*a}$, $Q_H$, and $Q_L$ were measured at a single point on the platform, and the source area of the EC measurements of $u_{*a}$, $Q_H$, and $Q_L$ ranges from 100 to 300 m along the wind direction over the lake (Mammarella et al., 2015). Thus, the values may not be representative for the whole lake. Wind speed and the resulting $u_{*a}$ over lakes surrounded by forests are lower in sheltered near-shore areas than in the central zones of the lakes (Markfort et al., 2010). Sheltering affects the spatial variation of wind speed especially in small lakes, such as Lake Kuivajärvi. Because $Q_H$

and $Q_L$ are dependent on wind speed over the lake, they may also be higher at the center of the lake than in near-shore areas. Also, the estimation of $u_{*a}$, $Q_H$, and $Q_L$ in the simulations was based on wind speed and other forcing data obtained from the single-point measurements, and the simulated values may have been overestimates of the spatial averages. However, despite the same measurement location, some disparities existed between the simulated and measured $Q_H$ and $Q_L$. The differences may be in part attributed to an underestimation of surface heat fluxes by the EC method, which was seen, for example, in a

study on energy balance over a small boreal lake by Nordbo et al. (2011). Considerable spatial variability may also occur in near-surface water $CO_2$ concentration in small, shallow boreal lakes (Natchimuthu et al., 2017), which may result in further discrepancies in the estimates on whole-lake $CO_2$ flux obtained on the basis of gas exchange models or by using a vertical, horizontally integrated lake model.

### 4.3   Factors influencing the epilimnetic $CO_2$ budget

The model parameter sets obtained through calibration of the MyLake C applications using different incorporated gas exchange models were notably different from each other, thus emphasizing different processes related to carbon cycling within the water column or to carbon exchange with the surrounding terrestrial ecosystem or the atmosphere. However, considering the main objective of the study, the simulation of near-surface $CO_2$ concentration and air–water $CO_2$ flux, the different outcomes of the calibration processes can be considered equally applicable as they give insight on the diversity of biogeochemical processes

that impact lacustrine $CO_2$ dynamics.

Phytoplankton is an significant factor in the lake $CO_2$ budget and the main driver of the diurnal variation of $CO_2$ concentration in Lake Kuivajärvi (Provenzale et al., 2018). In MyLake C, inorganic carbon is fixed by phytoplankton and carbon is stored in autochthonous organic matter within the water column or in bottom sediments until it is mineralized by bacteria. A relatively large portion of epilimnetic phytoplankton and dead autochthonous particulate organic matter sank from the

epilimnion into deeper layers in MI because of the small values of $m$ and $k_{POC,1}$. Production of $CO_2$ via degradation of phytoplankton-originated organic matter, as well as the release of bioavailable phosphorus in the epilimnion through mineralization of autochthonous organic matter, was also slow in MI because of a small $k_{DOC,1}$. As a result, the net production of $CO_2$ in the epilimnion was rather low in MI (Table 3) despite the relatively high simulated phytoplankton biomass. Overall, differences in total net $CO_2$ consumption by phytoplankton during the stratified period between GEMs were rather small despite



the large variation in the simulated phytoplankton biomasses because of the phosphorus limitation of photosynthesis in GEMs with a high phytoplankton biomass and because of the variation in the length of the active growing season between GEMs.

The simulated Chl $a$ concentrations were rather constant over the growing season with the exceptions of the substantial spring growth peaks in CC and TE. No data exist on Chl $a$ concentration in Lake Kuivajärvi in 2013, but the Chl $a$ concentration at

0–3 m was at its highest, 30–50 $\text{mg m}^{-3}$, in mid-July and decreased to a level of less than 2 $\text{mg m}^{-3}$ in late autumn in years 2011–2012 (Heiskanen et al., 2015). The epilimnetic Chl $a$ concentration is usually 3–5 $\text{mg m}^{-3}$ during the growing season with diatom-induced peaks under cool conditions in spring and autumn (Provenzale et al., 2018). Thus, the GEMs with low near-surface Chl $a$ concentrations, CC and TE, may have yielded better estimates of the overall phytoplankton biomass than HE and MI. The net consumption of $CO_2$ by phytoplankton was, however, not only related to the amount of phytoplankton

biomass. Nevertheless, none of the GEMs captured the supposed monthly variation of epilimnetic $CO_2$ concentration caused by the seasonal succession of phytoplankton.

A conspicuously high $C_{\text{DI,IN}}$ was needed to balance the high $CO_2$ efflux in the GEMs with a high $k$ (Table 1). The restriction of the scaling of the inflow DIC concentration to the open water season was a rough way to increase the gain of epilimnetic $CO_2$, and the summertime inflow $CO_2$ concentrations may have been unnaturally high especially in TE. However, the use of

$C_{\text{DI,IN}}$ can be thought as the inclusion of the input of $CO_2$ through groundwater seepage to the lake. In budget calculations, groundwater DIC load can be generally estimated by applying groundwater DIC flow as a percentage of stream DIC load (Chmiel et al., 2016). The amount of inflowing groundwater and its properties in Lake Kuivajärvi are unknown. However, in addition to inflow through minor inlet streams and surface runoff especially during snowmelt in spring, groundwater seepage may contribute somewhat to the total lake inflow volume because the measured total outflow volume over the year 2013 was

approximately double the inflowing volume via the main inlet stream. The $CO_2$ concentration in groundwater in southern Finland is around 700–900 $\text{mmol m}^{-3}$ (Lahermo et al., 1990), which is about tenfold higher than the estimated average inflow $CO_2$ concentration in Lake Kuivajärvi over the stratified period in 2013, 86 $\text{mmol m}^{-3}$, and well in line with the yearly average of groundwater $CO_2$ concentration near a boreal stream determined by Leith et al. (2015). Thus, also groundwater-derived $CO_2$ transport to the lake may affect the water column $CO_2$ concentration.

The effect of $CO_2$ inputs through minor inlets or groundwater may be supported by the fact that the simulated near-surface $CO_2$ concentration decreased too fast in all GEMs after ice-off in May 2013, that is, during a period when the snowmelt-induced flow in minor inlet streams may be substantial and when groundwater level is generally relatively high (Fig. 4). The simulated epilimnetic $CO_2$ sinks were rather small at that time because net $CO_2$ consumption by phytoplankton was low in cool water and because $CO_2$ efflux was relatively low because of a low air–water $CO_2$ concentration gradient. Labile, autochthonous DOC

was absent in the epilimnion in the simulations, and the degradation of allochthonous DOC was slow under the relatively cold conditions in May. Despite the measured inflow $CO_2$ concentration being approximately twice the simulated epilimnetic $CO_2$ concentration and the scaled inflow $CO_2$ concentrations and terrestrial $CO_2$ loads being even higher, the decline of epilimnetic $CO_2$ concentration was rapid in all GEMs. The high abundance of diatoms in Lake Kuivajärvi in spring may have resulted in a supply of easily degradable organic matter, but net primary production also consumed $CO_2$. Thus, substantial $CO_2$ loadings

through surface runoff, minor inlet streams, or groundwater seepage could have been plausible additional sources of epilimnetic



$CO_2$ in May, provided that the additional surface inflow was rich in $CO_2$. The impact of groundwater seepage is supported by a study on the carbon budget of a small boreal lake by Chmiel et al. (2016), in which the discrepancy between the estimates of gain and loss of inorganic carbon was explained by a possible underestimation of the impact of groundwater inflow.

## 5 Conclusions

We studied the applicability of four gas exchange models with different complexity incorporated into a vertical physicobiogeo-chemical lake model MyLake C to the simulation of air–water $CO_2$ exchange and water column $CO_2$ concentration in a humic boreal lake. The gas transfer velocities simulated using the simplest, wind-based gas exchange model by Cole and Caraco (1998), or CC, were best in accordance with the corresponding values calculated on the basis of direct in-lake measurements, whereas simulations with the other gas exchange models either overestimated (the models by Heiskanen et al. (2014) and

MacIntyre et al. (2010)) or underestimated (the model by Tedford et al. (2014)) the respective calculated gas transfer velocities because of discrepancies in the simulation of wind stress or daily effective surface heat flux.

None of the applied gas exchange models resulted in a highly improved simulation performance regarding water column $CO_2$ concentration or air–water $CO_2$ flux. On the contrary, the more complex gas exchange models, which include both wind-induced stress and heat-induced convection as the drivers of $CO_2$ exchange, yielded higher gas transfer velocities and thus

higher $CO_2$ fluxes in the simulations, which resulted in difficulties in obtaining sufficient gain of $CO_2$ in the water column to balance the loss to the atmosphere. In addition, the model with a daily time step was not always able to simulate the changes in near-surface $CO_2$ concentration and air–water $CO_2$ flux resulting from short-term physical processes, such as nighttime cooling or simultaneous surface heating and wind mixing. As a result, all the incorporated gas exchange models except for CC yielded notably too low summertime epilimnetic $CO_2$ concentrations in the simulations, which was also reflected by a

significant underestimation of $CO_2$ fluxes compared to the corresponding fluxes calculated from the calculated gas transfer velocities and measured air–water $CO_2$ concentration gradients. The daily $CO_2$ fluxes simulated with CC were closest to the corresponding calculated fluxes. The long and widely used CC was, however, shown to produce too low $CO_2$ flux estimates and its use was discouraged in an empirical gas exchange model intercomparison study by Erkkilä et al. (2018), whereas the more complex models yielded presumably more correct $CO_2$ fluxes. Nevertheless, it has to be noted that the comparison between

the gas exchange models is more complex in our modeling study than in Erkkilä et al. (2018) because of the interplay between the simulated $CO_2$ flux and water column $CO_2$ concentration.

The present model application was not highly adaptable to increased $CO_2$ effluxes. The extent of in-lake production of $CO_2$ is largely related to model structure, process descriptions, and the estimation of parameter values, whereas external $CO_2$ inputs are governed by the quality of hydrological forcing data. Thus, both further lake model development and improved estimates

of external loading are needed in order to enhance the predictive performance of model simulations. However, despite the challenges in using complex process-based models in the assessment of carbon cycling in lakes, modeling is an effective means to quantify underlying processes related to lacustrine $CO_2$ emissions and to study the development of lake ecosystems under changing conditions. The issues raised in our study concerning lacustrine carbon budgets can also be generalized to a




larger scale. If the application of advanced gas exchange models results in higher estimates of $CO_2$ emissions from boreal inland waters, a higher net terrestrial ecosystem production and an increased carbon flux from land to inland waters is required to close the regional carbon budget. Therefore, research on processes contributing to carbon cycling in boreal freshwaters and on the roles of different internal and external sources of $CO_2$, such as groundwater, in lakes is sorely needed.

5 *Code and data availability.* The MATLAB model code for the MyLake C application presented in this study is freely available at https://github.com/biogeochemistry/MyLake_C/tree/MyLake_C-gtsv. The automatic water column temperature and $CO_2$ concentration data, the SMEAR meteorological data, and the manual measurement data presented in this study are available in the AVAA - Open research data publishing platform (https://avaa.tdata.fi/web/smart/smear/download/). The metadata of these measurements are available via the ETSIN-service (https://etsin.avointiede.fi/).

10 *Author contributions.* PK, TV, AO, and TH conceived the idea and designed the modeling study. PK developed the model application, conducted the model simulations, and performed the analyses. IM, JH, and TV designed the field experiments. IM, JH, and HM participated in the field measurements. IM and KME participated in the automatic measurement data post-processing. PK prepared the manuscript with contributions from all co-authors.

*Competing interests.* The authors declare that they have no conflict of interest.

15 *Acknowledgements.* The work of Petri Kiuru was funded by grants awarded by the Kone Foundation and Maa- ja vesitekniikan tuki ry. The authors acknowledge the Academy of Finland Centre of Excellence (272041 and 118780), Academy Professor projects (1284701 and 1282842) and ICOS-Finland (281255) funded by Academy of Finland; the European Union Horizon 2020 project RINGO (730944); and the AtMath project funded by University of Helsinki. We thank Miitta Rantakari for the provision of the inflow data.



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





**Table 1.** Calibrated model parameters for the different versions of MyLake C application to Lake Kuivajärvi with different incorporated gas exchange models (HE: Heiskanen et al. (2014), CC: Cole and Caraco (1998), MI: MacIntyre et al. (2010), TE: Tedford et al. (2014)). The default parameter values were used as the means of the prior parameter distributions.

| | Default | HE | CC | MI | TE | Unit |
|---|---|---|---|---|---|---|
| $a_k$ | 3.92 | 0.27 | 0.45 | 0.39 | 1.18 | $\times 10^{-3}$ |
| $\beta_{DOC}$ | 2.85 | 2.94 | 3.47 | 3.22 | 2.75 | $\times 10^{-5} \text{ m}^2 \text{ mg}^{-2}$ |
| $C_{DI,IN}$ | 1.00 | 1.86 | 1.55 | 1.91 | 3.05 | - |
| $k_{DOC,1}$ | 0.80 | 5.71 | 1.11 | 0.46 | 9.01 | $\times 10^{-1} \text{ d}^{-1}$ |
| $k_{DOC,2}$ | 1.01 | 1.40 | 2.41 | 3.35 | 1.07 | $\times 10^{-2} \text{ d}^{-1}$ |
| $k_{POC,1}$ | 0.94 | 4.54 | 0.91 | 1.78 | 0.60 | $\times 10^{-1} \text{ d}^{-1}$ |
| $k_{POC,2}$ | 0.90 | 2.91 | 5.01 | 15.9 | 4.49 | $\times 10^{-2} \text{ d}^{-1}$ |
| $k_{POC,sed}$ | 2.53 | 4.11 | 2.43 | 2.84 | 3.72 | $\times 10^{-4} \text{ d}^{-1}$ |
| $m_{20}$ | 0.21 | 0.11 | 0.24 | 0.090 | 0.31 | $\text{d}^{-1}$ |
| $\mu'_{20}$ | 2.37 | 2.96 | 5.95 | 1.62 | 3.84 | $\text{d}^{-1}$ |
| $W_{str}$ | 0.29 | 0.33 | 0.35 | 0.35 | 0.24 | - |





**Table 2.** Total and monthly averages of simulated and calculated $CO_2$ fluxes ($\mu mol\ m^{-2}\ s^{-1}$) in May–October 2013 obtained with different gas exchange models. Only the days with available measurement data are included in the averaging of the simulated fluxes. Monthly values for June are excluded because measurement data were available only for 7 days.

| | May–October | | May | | July | |
|---|---|---|---|---|---|---|
| | Calc. | Sim. | Calc. | Sim. | Calc. | Sim. |
| Heiskanen | 0.38 | 0.31 | 0.79 | 0.41 | 0.37 | 0.34 |
| Cole & Caraco | 0.23 | 0.24 | 0.53 | 0.33 | 0.20 | 0.26 |
| MacIntyre | 0.45 | 0.29 | 0.97 | 0.52 | 0.44 | 0.26 |
| Tedford | 0.71 | 0.30 | 1.90 | 0.43 | 0.56 | 0.33 |
| | August | | September | | October | |
| | Calc. | Sim. | Calc. | Sim. | Calc. | Sim. |
| Heiskanen | 0.27 | 0.30 | 0.31 | 0.35 | 0.24 | 0.16 |
| Cole | 0.15 | 0.22 | 0.16 | 0.27 | 0.13 | 0.17 |
| MacIntyre | 0.33 | 0.22 | 0.35 | 0.32 | 0.25 | 0.16 |
| Tedford | 0.41 | 0.26 | 0.45 | 0.34 | 0.43 | 0.17 |





**Table 3.** Simulated $CO_2$ budgets (kg $CO_2$) for the epilimnion of Lake Kuivajärvi during summer stratification in 2013 and 2014 using different gas exchange models incorporated into MyLake C.

|  | Heiskanen | Cole & Caraco | MacIntyre | Tedford |
|---|---|---|---|---|
| **2013** |  |  |  |  |
| Net production | 52300 | 38700 | 40400 | 44800 |
| Change due to efflux[a] | −89900 | −72300 | −74500 | −89200 |
| Net external loading | 10800 | 8600 | 11000 | 18500 |
| Change due to epilimnion deepening | 16100 | 15100 | 15800 | 18800 |
| Change in epilimnetic storage | −10700 | −9900 | −7200 | −7100 |
| Duration (d) | 134 | 134 | 134 | 134 |
| **2014** |  |  |  |  |
| Net production | 38300 | 24900 | 25400 | 28700 |
| Change due to efflux[a] | −63600 | −42700 | −46600 | −57100 |
| Net external loading | 8300 | 6300 | 8100 | 12200 |
| Change due to epilimnion deepening | 17500 | 13100 | 14100 | 17400 |
| Change in epilimnetic storage | 500 | 1600 | 1000 | 1200 |
| Duration (d) | 107 | 101 | 101 | 101 |

[a] The change in water column $CO_2$ content due to $CO_2$ efflux was approximately 1 % lower than the amount of $CO_2$ evaded because of consequent equilibrium reactions in the carbonate system.



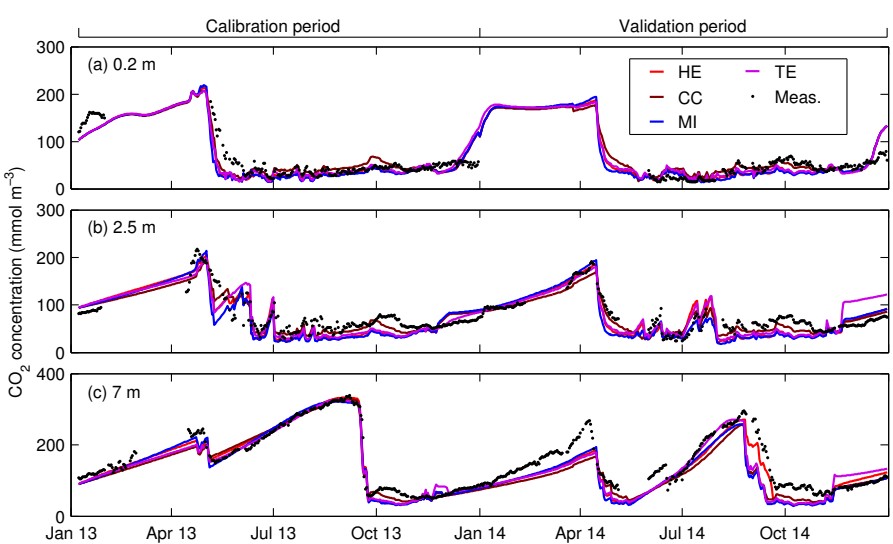

**Figure 1.** Simulation results for $CO_2$ concentration with each GEM (mmol m$^{-3}$) versus the daily averages of automatic high-frequency $CO_2$ concentration measurements at the depths of (a) 0.2 m, (b) 2.5 m, and (c) 7.0 m in Lake Kuivajärvi during the calibration year 2013 and the validation year 2014.





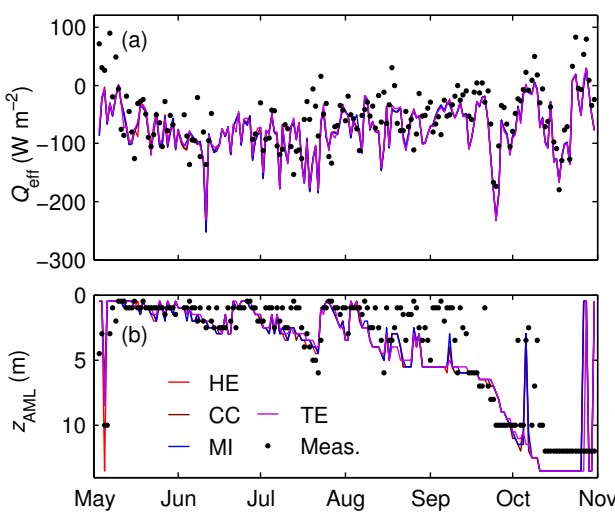

**Figure 2.** (a) Daily effective surface heat fluxes (W m$^{-2}$) simulated with each GEM and calculated on the basis of heat flux measurements and (b) simulated and empirically determined depths of the daily actively mixing layer (m) in Lake Kuivajärvi in May–October 2013.





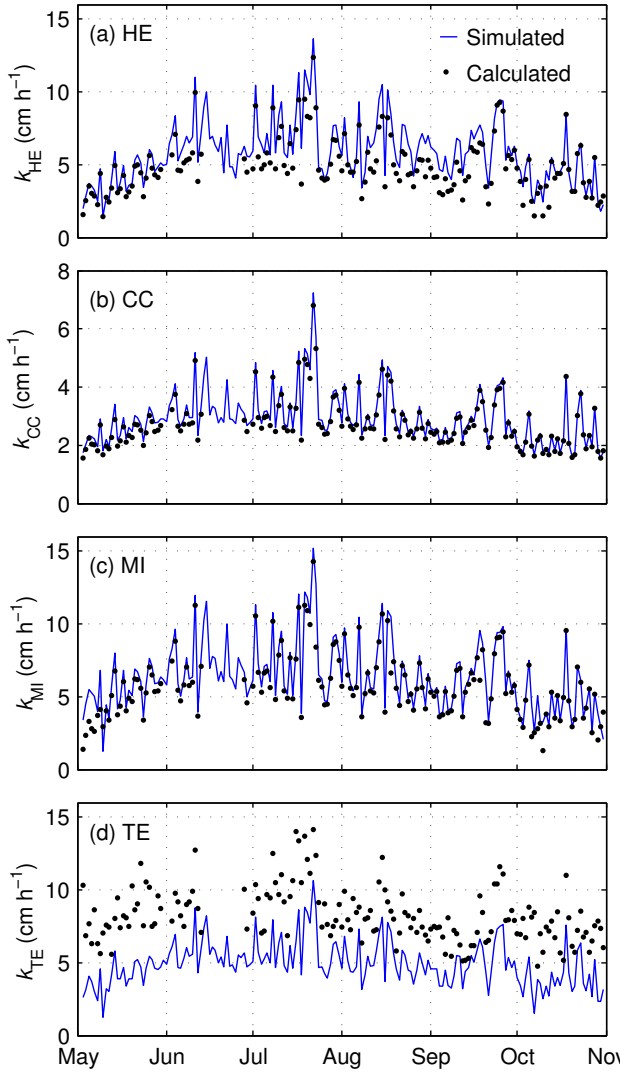

**Figure 3.** Simulated and calculated gas transfer velocities for $CO_2$ (cm h$^{-1}$) in Lake Kuivajärvi on 3 May–31 October 2013 obtained with the gas exchange models by (a) Heiskanen et al. (2014), (b) Cole and Caraco (1998), (c) MacIntyre et al. (2010), and (d) Tedford et al. (2014).





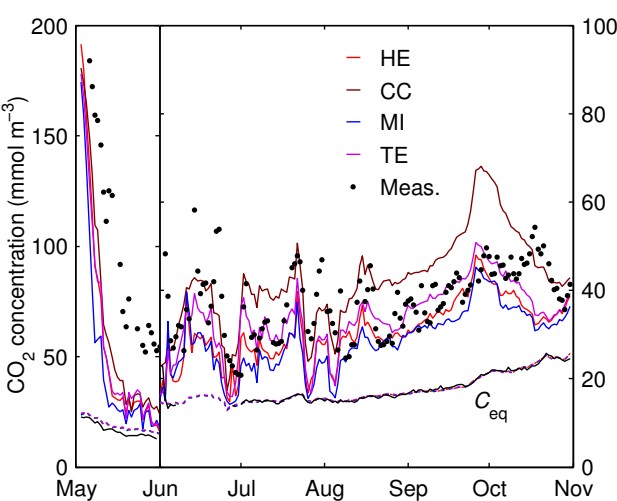

**Figure 4.** Simulated $CO_2$ concentrations ($\mathrm{mmol\ m^{-3}}$) in the surface layer (0–0.5 m) obtained with each GEM and the daily averages of the automatic measurements at 0.2 m in Lake Kuivajärvi in May–October 2013. Also shown are the atmospheric equilibrium concentrations of $CO_2$ ($C_{eq}$) obtained from the simulations (dotted colored lines) and calculated from the measured atmospheric $CO_2$ concentration and surface water temperature (solid black line). Note the different vertical scales in May and in June–October.



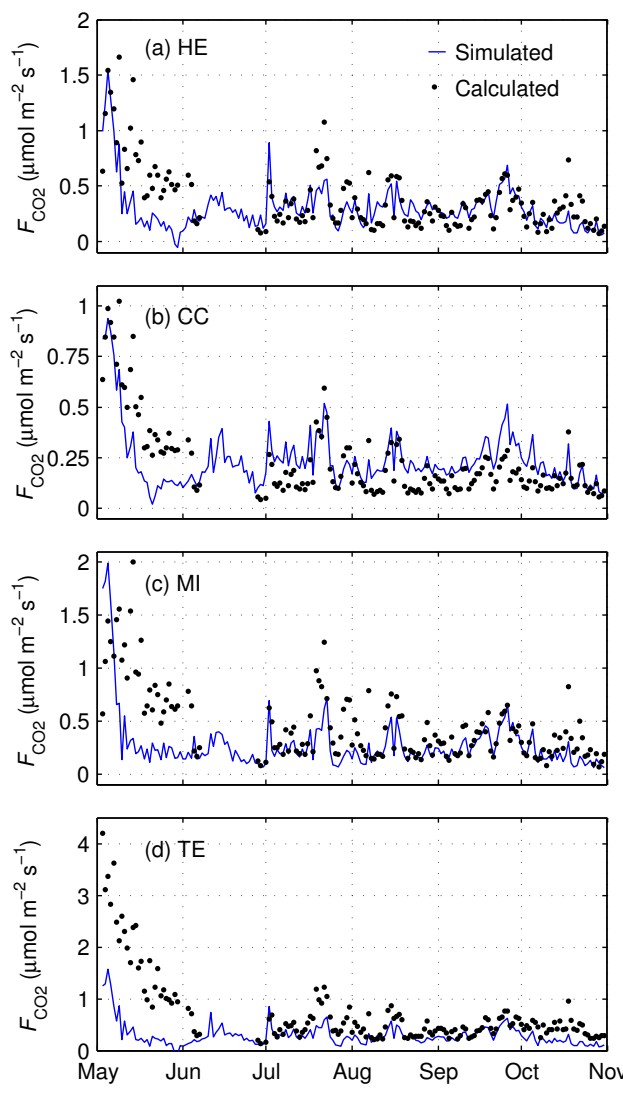

**Figure 5.** Simulated and calculated air–water $CO_2$ fluxes ($\mu$mol m$^{-2}$ s$^{-1}$) in Lake Kuivajärvi on 3 May–31 October 2013 obtained with the gas exchange models by (a) Heiskanen et al. (2014), (b) Cole and Caraco (1998), (c) MacIntyre et al. (2010), and (d) Tedford et al. (2014).





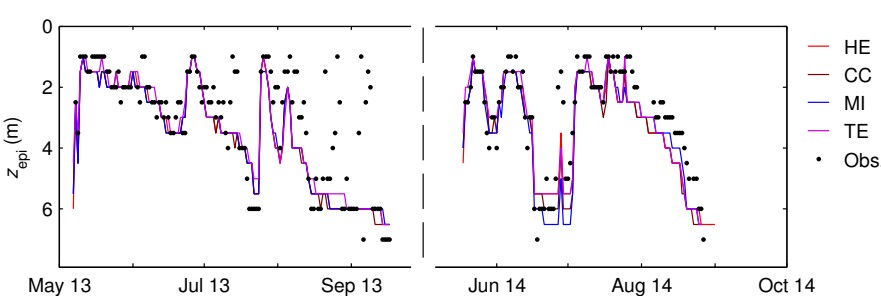

**Figure 6.** Simulated and observed depths of the epilimnion (m) in Lake Kuivajärvi during the continuous summer stratification in 2013 and 2014. The simulations were performed using each of the gas exchange models incorporated into MyLake C.