# Peer review of "Applicability and consequences of the integration of alternative models for CO2 transfer velocity into a process-based lake model"

_Biogeosciences, 2019_

## Referee Comment (RC1) · Anonymous Referee #1 · 8 May 2019

Overall, I find this to be useful work. Exercises such as this are not done as often as they should be. However, I am concerned about the model calibration and the overall message of the manuscript. I am unsure what overall message the authors are advocating. They do a comparison of the different parameterizations in a 1D model and leave it at that. The manuscript also has organizational issues which make it difficult to follow. To make this work more impactful, I suggest a section on modeling advice.

Please do not be discouraged by this review. I feel this work can be useful with some reorganization and reframing of the overall message. I very much look forward to reading a revised version.

Specific Comments

I am concerned about the model calibration. During the calibration step, the entire ecosystem is changed for each parametrization. I understand the calibration was intended to capture the surface $CO_2$ concentration. I would consider tuning the model capture some aspect of the ecosystem such as chlorophyll concentration.

It was also not clear why these specific parameterizations were chosen. Some rationale for choosing these specific parameterizations is needed. Admittedly, I am not familiar with most of these parameterizations, so the modeling community could benefit from a description of each. I suggest a section on "gas exchange parameterizations" where you start with a paragraph stating the gas exchange parameterizations and the parameters that go into them. I suggest putting all the parameters in a table with units. Additional sections can be descriptions of each parameterization and where it is currently being used (ie which models use them and which studies use them). Lastly, why wasn't Wanninkhof 1992 used in this comparison? Wann.1992 is the parametrization incorporated into ocean models such as the CESM and MITgcm. MITgcm has been used to in studies of the Great Lakes. Also, the chosen parameterizations are completely different from those used in marine environments (for example, Wrobel and Piskosub Ocean Sci., 12, 1091–1103, 2016 ). I can't think of any reason why there are different parameterizations for freshwater and marine systems.

I suggest a section providing modeling advice. Differences in gas transfer velocity and $CO_2$ flux using each method are mentions, but there is no consensus on which parametrization the community should be using. I also suggest highlighting more the impact the choice of these parametrizations has on global efflux from lakes.

Technical corrections

- Make it clear GEM stands for gas exchange model. It took me a minute to realize this.
- Add a table stating all the parameters with units used in each GEM - Figures 3 and 5 I suggest a cross plot off to the right with a list of summary statistics (correlation, bias,

RMSE, Nash-Sutcliffe efficiency, etc.) - I suggest a paragraph of modeling advice. How does this work advance modeling of the carbon cycle in lakes? - In the last paragraph of section 2.1.1 make it clear where the temperature dependent solubility comes into play. For this section I suggest looking at Wanninkhof et al. 2009 in annual review of marine science vol1:213-244. - In section 2.1.2 It is unclear where the approximation U10/U1.5=1.22 is used - in section 2.2.2. When you say the model was calibrated against daily averages of automatic CO2, does this simply mean the parameters in the model were tuned to match observed CO2 concentration? Please be clear about this. - In section 2.2.3 : please provide a rationale for this choice "Missing relative humidities were replaced by a value of 75 % in the calculation of the water-side friction velocity" - In section 2.2.4 : All the summary goodness-of-fit statistics (NS, B*, URMSE'*) can be displayed nicely in a target diagram. See Jolliff et al. 2009 "Summary diagrams for coupled hydrodynamic-ecosystem model skill assessment"

---

## Referee Comment (RC2) · Anonymous Referee #2 · 6 Jun 2019

The manuscript "The Applicability and consequences of the integration of alternative models for CO2 transfer velocity into a process-based lake model" of Petri Kiuru, Anne Ojala, Ivan Mammarella, Jouni Heiskanen, Kukka-Maaria Erkkilä, Heli Miettinen, Timo Vesala, and Timo Huttula is a interessting scientifc report about the performance of different gas exchange models for simulations of CO2 fluxes between lakes and atmosphere. The article represents the high scientific expertise of the finish research community. No doubt, the authors did a grandiose job. In my understanding, the article can be accepted after two minor improvements.

(1) The authors wrote on page 7, line 4 that the lake has a maximum width of only

0.3km. This raises the question whether the footprint of the EC measurements is really representative for the lake-atmosphere exchange. How did the authors approximate the width of the parabolic footprint? And how did the authors consider transversal advection, i.e., advection orthogonally to the mean flow (wind) direction?

(2) The authors discuss in section "4.2 Comparison to CO2 flux measurement" potential reasons for discrepancies between EC flux measurements and simulations results. Especially, they mentioned measurement errors and the spatial variability of governing parameters as major reasons. In my understanding, the authors are completely right with this statement. However, I would like to encourage the authors to provide quantitative support for this statement through a short error analysis.

---

## Author Comment (AC1) · 28 Jun 2019

bg-2019-95
**Author responses to comments of referee #1**

We would like to thank the referee for the effort and time he/she put in to review our manuscript. We are grateful for his/her careful and considered comments and will make every attempt to fully address these comments in the revised manuscript.

In the following list, the points raised by the referee are written in **bold** characters, whereas our responses are shown in blue characters.

**Overall, I find this to be useful work. Exercises such as this are not done as often as they should be. However, I am concerned about the model calibration and the overall message of the manuscript. I am unsure what overall message the authors are advocating. They do a comparison of the different parameterizations in a 1D model and leave it at that. The manuscript also has organizational issues which make it difficult to follow. To make this work more impactful, I suggest a section on modeling advice.**

**Please do not be discouraged by this review. I feel this work can be useful with some reorganization and reframing of the overall message. I very much look forward to reading a revised version.**

Our study is essentially a gas exchange model intercomparison study. The performance of the four parameterizations of $CO_2$ transfer velocity applied in this study has been estimated in prior studies by comparing the results to directly measured $CO_2$ fluxes and gas transfer velocities. In our study, we performed a corresponding comparison between the $CO_2$ transfer velocities and fluxes obtained through (a) lake model simulations and (b) calculation using measured $CO_2$ concentrations and other relevant variables. In addition, the aim of our study is to assess the capability of the lake model MyLake C to simulate lake inorganic carbon cycling especially under the conditions of high simulated $CO_2$ effluxes.

We will restructure the Methods section according to the Referee's suggestions and include further discussion on modeling advice and the overall conclusions of the study regarding the modeling of lake carbon cycling.

**Specific Comments**

**I am concerned about the model calibration. During the calibration step, the entire ecosystem is changed for each parametrization. I understand the calibration was intended to capture the surface CO2 concentration. I would consider tuning the model capture some aspect of the ecosystem such as chlorophyll concentration.**

We performed the model calibration against water column $CO_2$ concentration, and the aim of the calibration was indeed to optimize the simulated near-surface $CO_2$ concentration because air-water $CO_2$ exchange is governed by the air-water $CO_2$ concentration difference.

In the study, four different individual calibrations of the model application were performed. Because the lake model simulates a rather complex coupled physical-biogeochemical system and the statistical inference method used in the calibration tries to find an optimal parameter set using a relatively high number of free parameters, the individual parameter sets often tend to differ from each other. In other words, each statistical calibration yields a unique description of the lake carbon cycling. This is one of the reasons why the aim of the calibration procedure was not to try to reproduce the actual in-lake carbon cycling but rather to compare different possible ways to generate an optimal water column $CO_2$ concentration.

There are many possible drivers of in-lake $CO_2$ concentration variation, for example, phytoplankton processes, microbial degradation processes, and external loading of carbon species. Because phytoplankton is only one of the contributing factors and it is known that MyLake is not highly capable of simulating short-term phytoplankton dynamics correctly, different factors

were considered equal in the calibration. In addition, comprehensive data on water column $CO_2$ concentration were available, whereas chlorophyll $a$ measurements had not been performed in the lake during the study period.

**It was also not clear why these specific parameterizations were chosen. Some rationale for choosing these specific parameterizations is needed. Admittedly, I am not familiar with most of these parameterizations, so the modeling community could benefit from a description of each. I suggest a section on "gas exchange parameterizations" where you start with a paragraph stating the gas exchange parameterizations and the parameters that go into them. I suggest putting all the parameters in a table with units. Additional sections can be descriptions of each parameterization and where it is currently being used (ie which models use them and which studies use them). Lastly, why wasn't Wanninkhof 1992 used in this comparison? Wann.1992 is the parametrization incorporated into ocean models such as the CESM and MITgcm. MITgcm has been used to in studies of the Great Lakes. Also, the chosen parameterizations are completely different from those used in marine environments (for example, Wrobel and Piskosub Ocean Sci., 12, 1091–1103, 2016 ). I can't think of any reason why there are different parameterizations for freshwater and marine systems.**

We selected the four parameterizations, or gas exchange models, because the performance of these parameterizations has been assessed against direct $CO_2$ flux measurements in Lake Kuivajärvi in previous studies by Heiskanen et al. (2014), Mammarella et al. (2015), and Erkkilä et al. (2018). Consequently, also the simulations performed in our study could be indirectly compared with direct measurements in the study lake. We will include the reasoning in the text. The model by Wanninkhof (1992) is a wind-based parameterization, similarly to the parameterization by Cole and Caraco (1998). We chose to select the simple parameterization by Cole and Caraco (1998) to represent the parameterizations that are based only on wind speed in our study. Wind speed-based parameterizations have been shown to be inadequate in small lakes, and the use of more sophisticated parameterizations has been recommended (Heiskanen et al., 2014; Erkkilä et al., 2018).

Different parameterizations of air-water gas exchange for freshwater and marine systems are needed because the main drivers of near-surface turbulence are different in these systems. It has been shown that thermal convection is a larger source of mixed-layer turbulence than wind shear especially in lakes with a small surface area and a sheltered location (Read et al., 2012). (This may be the case also in oceanic regions with low to intermediate winds and strong insolation (see McGillis et al., 2004).) Thus, it has been suggested, for example, in the two aforementioned studies, that parameterizations relying only on wind speed may be insufficient under such conditions. Many boreal lakes are small in area, and also convective processes may have an essential role in air-water gas exchange in these lakes. By contrast, all the parameterizations in Wrobel and Piskozub (2016) are based solely on wind speed or include also gas transfer by bubbles. We will clarify the limited applicability of wind-based parameterizations in the Introduction:

"Buoyancy flux is relatively more important in small, wind-sheltered lakes, and parameterizations of the gas transfer velocity that are based solely on wind speed may not be applicable under such conditions (Read et al, 2012)."

We will revise the manuscript to include a more detailed description of the applied parameterizations and related parameters according to the Referee's suggestions. The main parameters will be tabulated and their descriptions will be given, followed by the descriptions of the gas exchange models and their prior usage. However, as far as we know, the models by MacIntyre et al. (2010) and Tedford et al. (2014) have not been widely used in other studies, let alone having been integrated into biogeochemical models. Furthermore, we consider that the model

by Cole and Caraco is so well-established and widely known in the field that there is no need to review its usage further.

**I suggest a section providing modeling advice. Differences in gas transfer velocity and CO2 flux using each method are mentions, but there is no consensus on which parametrization the community should be using. I also suggest highlighting more the impact the choice of these parametrizations has on global efflux from lakes.**

We have concluded that it is not a trivial task to judge which parameterization is most suitable for integration into MyLake C. None of the four model versions with different gas transfer velocity parameterizations surpassed the other ones in the study because of the complex interplay between the near-surface water $CO_2$ concentration and air-water $CO_2$ flux in the simulations. However, many experimental studies have shown that traditional, wind-based parameterizations often yield too low fluxes when compared to estimates based on direct measurements. Thus, we find that it is recommended to strive to use the more sophisticated gas exchange models provided that the lake biogeochemical model can be made adaptable to higher $CO_2$ losses and that the parameters related to the convection-based parameterizations can be simulated correctly.

We will include a more detailed discussion on our recommendations on the selection of the gas exchange model. We will also state more clearly also in the Discussion that the estimates of global gas efflux will be higher if more correct gas exchange parameterizations are used.

**Technical corrections**
**- Make it clear GEM stands for gas exchange model. It took me a minute to realize this.**
GEM is actually intended to stand for an individual MyLake C version that uses one of the four different gas exchange models/parameterizations. When the actual model/parameterization (that is, the formula) is discussed, the phrase "gas exchange model" is used in the text.

We will clarify the usage of the abbreviation GEM in the manuscript. The abbreviation has been defined at its first occurrence in section 2.2.4. We will also repeat the definition of the abbreviation at the beginning of the Results and Discussion sections.

"Even though the differences between the formulations of the gas exchange models incorporated into MyLake C are rather notable, the resultant $CO_2$ concentrations did not differ substantially between the GEMs, that is, between the simulations with the MyLake C versions using different gas exchange models (Fig. 1)."

"There was less variation between the air–water $CO_2$ fluxes simulated with different GEMs, that is, simulated with the MyLake C versions using different gas exchange models, than between the $CO_2$ fluxes calculated with the corresponding different gas exchange models on the basis on measured surface heat fluxes and air–water $CO_2$ concentration gradients (Table 3)."

**- Add a table stating all the parameters with units used in each GEM**
We will add the table as suggested.

**- Figures 3 and 5 I suggest a cross plot off to the right with a list of summary statistics (correlation, bias, RMSE, Nash-Sutcliffe efficiency, etc.)**
We will include a cross plot containing also some summary statistics along with each subplot in Figures 3 and 5.

**- I suggest a paragraph of modeling advice. How does this work advance modeling of the carbon cycle in lakes?**
We believe that this work addresses the issue of the selection of the gas transfer parameterization to be used in biogeochemical lake models. It is clear – on the basis of many previous

experimental studies – that traditional wind-based gas exchange parameterizations tend to give too low gas fluxes and that more advanced models with higher flux estimates should be used. However, our study suggests that it is not a straightforward task to simply use a better gas exchange model in a lake model because it may bring about difficulties in the simulation of in-lake carbon cycling, particularly in the generation of sufficient gain of $CO_2$ in the water column. Thus, we conclude that further development related to the mathematical description of in-lake carbon processes and to the modeling or other kinds of estimation of external inorganic and organic carbon loading are still needed.

Please see also our answer to the prior Specific Comment that is related to this comment.

**- In the last paragraph of section 2.1.1 make it clear where the temperature dependent solubility comes into play. For this section I suggest looking at Wanninkhof et al. 2009 in annual review of marine science vol1:213-244.**

We agree that there are different ways to describe the air-water concentration difference at equilibrium by using different solubility coefficients (for example, Henry's law solubility constant, Ostwald solubility coefficient, and Bunsen solubility coefficient). We will clarify that we meant the temperature dependence of the Henry's law solubility constant $K_H$:

"[...] where $K_H$ is the temperature-dependent aqueous-phase solubility (also known as the Henry's law constant) of $CO_2$ at surface water temperature, [...]"

"[...] and the temperature dependence of the aqueous-phase solubility $K_H$ is calculated according to Weiss (1974)."

**- In section 2.1.2 It is unclear where the approximation U10/U1.5=1.22 is used**

The gas exchange models by Cole and Caraco (1998) and MacIntyre et al. (2010) use the wind speed at 10 m as input, whereas the model by Tedford et al. (2014) uses wind speed at 1.5 m. In the calculations, wind speed measurements performed at 1.5 m height (which is, by chance, also the height used in the model by Tedford et al. (2014)) are used, and the conversion between wind speeds at 1.5 and 10 m is performed using the approximation $U_{10}/U_{1.5} = 1.22$. The measurement height used in this study, 1.5 m, is stated later on in the text in section 2.2.2; however, the conversion factor is included in the MyLake C model code and is thus not only a study-specific value. As the structure of the manuscript will be modified, we will move the statement to an appropriate location and restate the statement in the description of model assessment data in section 2.2.3.

**- in section 2.2.2. When you say the model was calibrated against daily averages of automatic CO2, does this simply mean the parameters in the model were tuned to match observed CO2 concentration? Please be clear about this.**

This is exactly what we meant. The calibration procedure is explained in more detail later on in the text, in section 2.2.4. In this section (2.2.2), the measurements used in the calibration procedure are described. The calibration method was statistical, and it would be imprecise to merely state that the model parameters were tuned to match the observed $CO_2$ concentration, to minimize the root-mean-square error (RMSE) between the simulated and measured concentrations, or to improve the fit between the simulated and calibrated values.

We will clarify the statement as follows: "The model was calibrated against the daily averages of the automatic high-frequency $CO_2$ concentration measurements: an optimal set of selected model parameters were estimated so that the simulated $CO_2$ concentration time series matched the corresponding measured $CO_2$ concentration time series as well as possible. The estimation was performed using a statistical inference algorithm."

**- In section 2.2.3 : please provide a rationale for this choice "Missing relative humidities**

**were replaced by a value of 75 % in the calculation of the water-side friction velocity"**

Many half-hour values of $k_{TE}$ could not be calculated because of missing data on surface heat fluxes (described in section 2.2.3) or other variables, and the omission of periods with missing relative humidity would have further decreased the number of calculated $k_{TE}$ values. Thus, we chose to approximate the missing values of relative humidity.

Furthermore, relative humidity has a very small effect on the gas transfer velocity calculated with the parameterization by Tedford et al. (2014) ($k_{TE}$), and it is not included in the other parameterizations. The Air-Sea Toolbox utilized by MyLake includes a formula that calculates air density $\rho_a$ on the basis of air temperature, relative humidity, and air pressure. The variation of air density at different relative humidities compared to the value of 70 % is, at maximum, of the order of 0.1 %. Because $k_{TE}$ is proportional to $\rho_a^{3/8}$, the corresponding variation of the gas transfer velocity is even smaller. However, we chose to include the statement on the relative humidity for completeness.

The mean value of the SMEAR measurements of relative humidity during the period May–October 2013 was 72 %. Platform measurements of relative humidity were relatively well applicable during the period May–August 2013. During May–August, the average values for the SMEAR and platform measurements of relative humidity were 66 % and 68 %, respectively. Thus, the average relative humidity can be assumed to be slightly higher over the lake than at the SMEAR station. Consequently, we find that a value of 75 % is a relatively good as a rough approximation for the average relative humidity at the platform during May–October. We will explain the choice in the text:

"In the calculation of the water-side friction velocity, missing relative humidities were replaced by a value of 75 %, which is close to the average of the SMEAR II measurements of relative humidity in May–October 2013, 72 %. The corresponding averages over the period May–August 2013, for which platform measurements were rather well applicable, were 66 % and 68 % for the SMEAR II and platform measurements, respectively. Thus, the relative humidity can be assumed to have been slightly higher over the lake than at the SMEAR II station."

**- In section 2.2.4 : All the summary goodness-of-fit statistics (NS, B\*, URMSE'\*) can be displayed nicely in a target diagram. See Jolliff et al. 2009 "Summary diagrams for coupled hydrodynamic-ecosystem model skill assessment"**

All the statistics are presented in tables in the Supplement. Because the manuscript already contains a rather high amount of figures, we consider that additional target graphs would not provide much additional value to the manuscript.

References

Cole, J. J. and Caraco, N. F. (1998), Atmospheric exchange of carbon dioxide in a low-wind oligotrophic lake measured by the addition of SF$_6$, Limnol. Oceanogr., 43, 647–656, doi:10.4319/lo.1998.43.4.0647.

Erkkilä, K.-M., et al. (2018), Methane and carbon dioxide fluxes over a lake: Comparison between eddy covariance, floating chambers and boundary layer method, Biogeosciences, 15, 429–445, doi:10.5194/bg-15-429-2018.

Heiskanen, J. J., et al. (2014), Effects of cooling and internal wave motions on gas transfer coefficients in a boreal lake, Tellus B, 66, 22 827, doi:10.3402/tellusb.v66.22827.

MacIntyre, S., et al. (2010), Buoyancy flux, turbulence, and the gas transfer coefficient in a stratified lake, Geophys. Res. Lett., 37, L24 604, doi:10.1029/2010GL044164.

Mammarella, I., et al. (2015), Carbon dioxide and energy fluxes over a small boreal lake in Southern Finland, J. Geophys. Res.-Biogeosci., 120, 1296–1314, doi:10.1002/2014JG002873.

McGillis, W. R., et al. (2004), Air-sea CO2 exchange in the equatorial Pacific, J. Geophys. Res., 109, C08S02, doi:10.1029/2003JC002256.

Read, J. S., et al. (2012), Lake-size dependency of wind shear and convection as controls on gas exchange, Geophys. Res. Lett., 39, L09 405, doi:10.1029/2012GL051886.

Tedford, E. W., et al. (2014), Similarity scaling of turbulence in a temperate lake during fall cooling, J. Geophys. Res.-Oceans, 119, 4689–4713, doi:10.1002/2014JC010135.

---

## Author Comment (AC2) · 28 Jun 2019

bg-2019-95
**Author responses to comments of referee #2**

We would like to thank the referee for the effort and time he/she put in to review our manuscript. We are grateful for his/her careful and considered comments and will make every attempt to fully address these comments in the revised manuscript.

In the following list, the points raised by the referee are written in **bold** characters, whereas our responses are shown in blue characters.

**The manuscript "The Applicability and consequences of the integration of alternative models for CO2 transfer velocity into a process-based lake model" of Petri Kiuru, Anne Ojala, Ivan Mammarella, Jouni Heiskanen, Kukka-Maaria Erkkilä, Heli Miettinen, Timo Vesala, and Timo Huttula is a interessting scientifc report about the performance of different gas exchange models for simulations of CO2 fluxes between lakes and atmosphere. The article represents the high scientific expertise of the finish research community. No doubt, the authors did a grandiose job. In my understanding, the article can be accepted after two minor improvements.**

**(1) The authors wrote on page 7, line 4 that the lake has a maximum width of only 0.3km. This raises the question whether the footprint of the EC measurements is really representative for the lake-atmosphere exchange. How did the authors approximate the width of the parabolic footprint? And how did the authors consider transversal advection, i.e., advection orthogonally to the mean flow (wind) direction?**

The estimation of the flux footprint distribution functions was made using the model by Kormann and Meixner (2001). The average footprint contributing to 80 % of the flux ranges from 100 m up to about 300 m from the measurement platform depending on atmospheric stability conditions as described in Mammarella et al. (2015). However, the simple footprint model may have overestimated the footprint because it does not take into account the additional turbulence generated by the surrounding forest. Nevertheless, it is justified to assume that the source area of the measured fluxes was on the lake surface because only the measurements during the periods when the wind was blowing along the lake were used in the analysis.

The wind is channeled along the lake for most of the time. When the wind is blowing along the lake, the footprints are within the lake fetch. Transversal wind directions were filtered out in the data used in the study. Typically, 15 % of the flux data are excluded from the analysis, when the wind is not blowing along the lake (the excluded wind directions are 350°–130° and 180°–320°). However, in calm nights some air can be transversally advected even if the wind is along the lake. In principle, the standard quality checking (described in detail in Mammarella et al. (2015)) removes the data contaminated by advection. Although the advection may still affect the concentrations and temperatures, the covariances with wind, that is, the eddy fluxes, are somewhat immune to advective effects.

We will add discussion on the foregoing issues in section 2.2.3.

"The estimation of the flux footprint distribution functions was made using the model by Kormann and Meixner (2001). The average footprint contributing to 80 % of the fluxes varies from 100 m up to about 300 m from the measurement platform depending on atmospheric stability conditions as described in Mammarella et al. (2015). Only wind directions along the lake (130°–180° and 320°–350°) were included to ensure that heat fluxes from the surrounding land were excluded. Furthermore, possible remaining effects of transversal advection during calm nights were removed through EC quality screening."

**(2) The authors discuss in section "4.2 Comparison to CO2 flux measurement" potential reasons for discrepancies between EC flux measurements and simulations results. Especially, they mentioned measurement errors and the spatial variability of governing parameters as major reasons. In my understanding, the authors are completely right with this statement. However, I would like to encourage the authors to provide quantitative support for this statement through a short error analysis.**

Estimates of the random uncertainty of EC fluxes on Lake Kuivajärvi for the years 2010 and 2011 have been studied in detail in Mammarella et al. (2015). On average, the estimated total relative random error was around 10 % for both sensible and latent heat fluxes. The estimated relative $CO_2$ flux random error was approximately double as large as that of energy fluxes, 20 % of measured fluxes, which is a typical value for EC $CO_2$ flux reported also in other types of ecosystems.

We will include some quantitative error analysis, related to both the underestimation of surface heat fluxes and the random measurement error, in the text:

"[...] The differences may be in part attributed to an underestimation of surface heat fluxes by the EC method, which was seen, for example, in a study on energy balance over a small boreal lake by Nordbo et al. (2011) and also in Mammarella et al. (2015). The sum of the measured EC heat fluxes in Lake Kuivajärvi was on average 83 % and 79 % of available energy in 2010 and 2011, respectively, in Mammarella et al. (2015). In addition, the total relative random error of the EC measurements is generally around 10 % for both sensible heat flux and latent heat flux as estimated in Mammarella et al. (2015). [...]"

References

Kormann, R. and Meixner, F. X. (2001), An analytical footprint model for non-neutral stratification, Bound.-Lay. Meteorol., 99, 2, 207–224, doi:10.1023/A:1018991015119.

Mammarella, I., et al. (2015), Carbon dioxide and energy fluxes over a small boreal lake in Southern Finland, J. Geophys. Res.-Biogeosci., 120, 1296–1314, doi:10.1002/2014JG002873.

---

## Author Response (AR1)

bg-2019-95
**Author responses to comments**

We would like to thank the editor and the two referees for the effort and time they put in to review our manuscript. We are grateful for their careful and considered comments and have made every attempt to fully address these comments in the revised manuscript.

In the following list, the points raised by the referees are written in **bold** characters, whereas our responses are shown in blue characters and the text excerpts indicating the corresponding changes made to the manuscript are shown in orange characters. The line numbers within our responses correspond to those in the revised manuscript file.

A separate list of major structural changes and figure/table revisions is included after the point-by-point responses to the referee comments.

**Referee #1**

**Overall, I find this to be useful work. Exercises such as this are not done as often as they should be. However, I am concerned about the model calibration and the overall message of the manuscript. I am unsure what overall message the authors are advocating. They do a comparison of the different parameterizations in a 1D model and leave it at that. The manuscript also has organizational issues which make it difficult to follow. To make this work more impactful, I suggest a section on modeling advice.**

**Please do not be discouraged by this review. I feel this work can be useful with some reorganization and reframing of the overall message. I very much look forward to reading a revised version.**

Our study is essentially a gas exchange model intercomparison study. The performance of the four parameterizations of $CO_2$ transfer velocity applied in this study has been estimated in prior studies by comparing the results to directly measured $CO_2$ fluxes and gas transfer velocities. In our study, we performed a corresponding comparison between the $CO_2$ transfer velocities and fluxes obtained through (a) lake model simulations and (b) calculation using measured $CO_2$ concentrations and other relevant variables. In addition, the aim of our study is to assess the capability of the lake model MyLake C to simulate lake inorganic carbon cycling especially under the conditions of high simulated $CO_2$ effluxes.

We have restructured the Methods section according to the Referee's suggestions and included further discussion on modeling advice and the overall conclusions of the study regarding the modeling of lake carbon cycling.

**Specific Comments**
**I am concerned about the model calibration. During the calibration step, the entire ecosystem is changed for each parametrization. I understand the calibration was intended to capture the surface CO2 concentration. I would consider tuning the model capture some aspect of the ecosystem such as chlorophyll concentration.**

We performed the model calibration against water column $CO_2$ concentration, and the aim of the calibration was indeed to optimize the simulated near-surface $CO_2$ concentration because air-water $CO_2$ exchange is governed by the air-water $CO_2$ concentration difference.

In the study, four different individual calibrations of the model application were performed. Because the lake model simulates a rather complex coupled physical-biogeochemical system and the statistical inference method used in the calibration tries to find an optimal parameter set using a relatively high number of free parameters, the individual parameter sets often tend to differ from each other. In other words, each statistical calibration yields a unique description

of the lake carbon cycling. This is one of the reasons why the aim of the calibration procedure was not to try to reproduce the actual in-lake carbon cycling but rather to compare different possible ways to generate an optimal water column $CO_2$ concentration.

There are many possible drivers of in-lake $CO_2$ concentration variation, for example, phytoplankton processes, microbial degradation processes, and external loading of carbon species. Because phytoplankton is only one of the contributing factors and it is known that MyLake is not highly capable of simulating short-term phytoplankton dynamics correctly, different factors were considered equal in the calibration. In addition, comprehensive data on water column $CO_2$ concentration were available, whereas chlorophyll *a* measurements had not been performed in the lake during the study period.

**It was also not clear why these specific parameterizations were chosen. Some rationale for choosing these specific parameterizations is needed. Admittedly, I am not familiar with most of these parameterizations, so the modeling community could benefit from a description of each. I suggest a section on "gas exchange parameterizations" where you start with a paragraph stating the gas exchange parameterizations and the parameters that go into them. I suggest putting all the parameters in a table with units. Additional sections can be descriptions of each parameterization and where it is currently being used (ie which models use them and which studies use them). Lastly, why wasn't Wanninkhof 1992 used in this comparison? Wann.1992 is the parametrization incorporated into ocean models such as the CESM and MITgcm. MITgcm has been used to in studies of the Great Lakes. Also, the chosen parameterizations are completely different from those used in marine environments (for example, Wrobel and Piskosub Ocean Sci., 12, 1091–1103, 2016 ). I can't think of any reason why there are different parameterizations for freshwater and marine systems.**

We selected the four parameterizations, or gas exchange models, because the performance of these parameterizations has been assessed against direct $CO_2$ flux measurements in Lake Kuivajärvi in previous studies by Heiskanen et al. (2014), Mammarella et al. (2015), and Erkkilä et al. (2018). Consequently, also the simulations performed in our study could be indirectly compared with direct measurements in the study lake. We have include the reasoning in the text.

P3 L24–27: "The four gas exchange models were selected because their performance in estimating air–water $CO_2$ fluxes in a small boreal lake has been extensively assessed in previous studies by Heiskanen et al. (2014), Mammarella et al. (2015), and Erkkilä et al. (2018) by comparing the calculated fluxes with direct $CO_2$ flux measurements."

The model by Wanninkhof (1992) is a wind-based parameterization, similarly to the parameterization by Cole and Caraco (1998). We chose to select the simple parameterization by Cole and Caraco (1998) to represent the parameterizations that are based only on wind speed in our study. Wind speed-based parameterizations have been shown to be inadequate in small lakes, and the use of more sophisticated parameterizations has been recommended (Heiskanen et al., 2014; Erkkilä et al., 2018).

Different parameterizations of air-water gas exchange for freshwater and marine systems are needed because the main drivers of near-surface turbulence are different in these systems. It has been shown that thermal convection is a larger source of mixed-layer turbulence than wind shear especially in lakes with a small surface area and a sheltered location (Read et al., 2012). (This may be the case also in oceanic regions with low to intermediate winds and strong insolation (see McGillis et al., 2004).) Thus, it has been suggested, for example, in the two aforementioned studies, that parameterizations relying only on wind speed may be insufficient under such conditions. Many boreal lakes are small in area, and also convective processes may have an essential role in air-water gas exchange in these lakes. By contrast, all the parameterizations in Wrobel and Piskozub (2016) are based solely on wind speed or include also gas

transfer by bubbles. We have clarified the limited applicability of wind-based parameterizations in the Introduction:

P2 L17–18: "Buoyancy flux is relatively more important in small, wind-sheltered lakes, and parameterizations of the gas transfer velocity that are based solely on wind speed may not be applicable under such conditions (Read et al, 2012)."

We have revised the manuscript to include a more detailed description of the applied parameterizations and related parameters according to the Referee's suggestions. The main parameters are tabulated in Table 1 and their descriptions are given, followed by the descriptions of the gas exchange models and their prior usage. However, as far as we know, the models by MacIntyre et al. (2010) and Tedford et al. (2014) have not been widely used in other studies, let alone having been integrated into biogeochemical models. Furthermore, we consider that the model by Cole and Caraco is so well-established and widely known in the field that there was no need to review its usage further.

**I suggest a section providing modeling advice. Differences in gas transfer velocity and CO2 flux using each method are mentions, but there is no consensus on which parametrization the community should be using. I also suggest highlighting more the impact the choice of these parametrizations has on global efflux from lakes.**

We have concluded that it is not a trivial task to judge which parameterization is most suitable for integration into MyLake C. None of the four model versions with different gas transfer velocity parameterizations surpassed the other ones in the study because of the complex interplay between the near-surface water $CO_2$ concentration and air-water $CO_2$ flux in the simulations. However, many experimental studies have shown that traditional, wind-based parameterizations often yield too low fluxes when compared to estimates based on direct measurements. Thus, we find that it is recommended to strive to use the more sophisticated gas exchange models provided that the lake biogeochemical model can be made adaptable to higher $CO_2$ losses and that the parameters related to the convection-based parameterizations can be simulated correctly.

We have added section 4.4, which contains a more detailed discussion on our recommendations on the selection of the gas exchange model. We have also stated more clearly also in the Discussion that the estimates of global gas efflux will be higher if more correct gas exchange parameterizations are used.

P22 L31–P23 L2: "The issues raised in our study concerning lacustrine carbon budgets can also be generalized to a larger scale. The application of advanced gas exchange models has been shown to lead to increased estimates of $CO_2$ emissions from boreal inland waters. Thus, higher estimates of net terrestrial ecosystem production and carbon flux from land to inland waters are required to close the regional carbon budget. Also, the use of advanced, possibly more correct gas exchange models in the assessment of global gas efflux from freshwaters may result in higher estimates of the impact of freshwater ecosystems on global carbon cycling."

**Technical corrections**
**- Make it clear GEM stands for gas exchange model. It took me a minute to realize this.**

GEM is actually intended to stand for an individual MyLake C version that uses one of the four different gas exchange models/parameterizations. When the actual model/parameterization (that is, the formula) is discussed, the phrase "gas exchange model" is used in the text.

We have clarified the usage of the abbreviation GEM in the manuscript. The abbreviation has originally been defined at its first occurrence in section 2.2.4. We have also repeated the definition of the abbreviation at the beginning of the Results and Discussion sections.

P12 L3–5: "Even though the differences between the formulations of the gas exchange models incorporated into MyLake C are rather notable, the resultant $CO_2$ concentrations did not differ

substantially between the GEMs, that is, between the simulations with the MyLake C versions using different gas exchange models (Fig. 1)."

P17 L15–17: "There was less variation between the air–water $CO_2$ fluxes simulated with different GEMs, that is, simulated with the MyLake C versions using different gas exchange models, than between the $CO_2$ fluxes calculated with the corresponding different gas exchange models on the basis on measured surface heat fluxes and air–water $CO_2$ concentration gradients (Table 3)."

**- Add a table stating all the parameters with units used in each GEM**
We have added the table (Table 1) as suggested.

**- Figures 3 and 5 I suggest a cross plot off to the right with a list of summary statistics (correlation, bias, RMSE, Nash-Sutcliffe efficiency, etc.)**
We have included a cross plot containing also some summary statistics along with each subplot in Figures 3 and 5.

**- I suggest a paragraph of modeling advice. How does this work advance modeling of the carbon cycle in lakes?**
We believe that this work addresses the issue of the selection of the gas transfer parameterization to be used in biogeochemical lake models. It is clear – on the basis of many previous experimental studies – that traditional wind-based gas exchange parameterizations tend to give too low gas fluxes and that more advanced models with higher flux estimates should be used. However, our study suggests that it is not a straightforward task to simply use a better gas exchange model in a lake model because it may bring about difficulties in the simulation of in-lake carbon cycling, particularly in the generation of sufficient gain of $CO_2$ in the water column. Thus, we conclude that further development related to the mathematical description of in-lake carbon processes and to the modeling or other means of estimation of external inorganic and organic carbon loading are still needed.
Please see also our answer to the prior Specific Comment that is related to this comment.

**- In the last paragraph of section 2.1.1 make it clear where the temperature dependent solubility comes into play. For this section I suggest looking at Wanninkhof et al. 2009 in annual review of marine science vol1:213-244.**
We agree that there are different ways to describe the air-water concentration difference at equilibrium by using different solubility coefficients (for example, Henry's law solubility constant, Ostwald solubility coefficient, and Bunsen solubility coefficient). We have clarified that we meant the temperature dependence of the Henry's law solubility constant $K_H$:

P4 L11–12: "[. . . ] where $K_H$ is the temperature-dependent aqueous-phase solubility (also known as the Henry's law constant) of $CO_2$ at surface water temperature, [. . . ]"

P7 L26–27: "[. . . ] and the temperature dependence of the aqueous-phase solubility $K_H$ is calculated according to Weiss (1974)."

**- In section 2.1.2 It is unclear where the approximation U10/U1.5=1.22 is used**
The gas exchange models by Cole and Caraco (1998) and MacIntyre et al. (2010) use the wind speed at 10 m as input, whereas the model by Tedford et al. (2014) uses wind speed at 1.5 m. In the calculations, wind speed measurements performed at 1.5 m height (which is, by chance, also the height used in the model by Tedford et al. (2014)) are used, and the conversion between wind speeds at 1.5 and 10 m is performed using the approximation $U_{10}/U_{1.5} = 1.22$. The measurement height used in this study, 1.5 m, is stated later on in the text in section 2.2.2; however, the conversion factor is included in the MyLake C model code and is thus not only

a study-specific value. As the structure of the manuscript has been modified, we have moved the statement to the end of section 2.1.3 and restated the statement in the description of model assessment data in section 2.2.3.

P7 L33: The approximation $U_{10}/U_{1.5} = 1.22$ is used for the wind speed at different heights.

P10 L20–21: [. . . ] As in MyLake C, the approximation $U_{10}/U_{1.5} = 1.22$ was used in the calculations.

**- in section 2.2.2. When you say the model was calibrated against daily averages of automatic CO2, does this simply mean the parameters in the model were tuned to match observed CO2 concentration? Please be clear about this.**

This is exactly what we meant. The calibration procedure is explained in more detail later on in the text, in section 2.2.4. In this section (2.2.2), the measurements used in the calibration procedure are described. The calibration method was statistical, and it would be imprecise to merely state that the model parameters were tuned to match the observed $CO_2$ concentration, to minimize the root-mean-square error (RMSE) between the simulated and measured concentrations, or to improve the fit between the simulated and calibrated values.

We have clarified the statement as follows:

P9 L5–8: "The model was calibrated against the daily averages of the automatic high-frequency $CO_2$ concentration measurements: an optimal set of selected model parameters were estimated so that the simulated $CO_2$ concentration time series matched the corresponding measured $CO_2$ concentration time series as well as possible. The estimation was performed using a statistical inference algorithm."

**- In section 2.2.3 : please provide a rationale for this choice "Missing relative humidities were replaced by a value of 75 % in the calculation of the water-side friction velocity"**

Many half-hour values of $k_{TE}$ could not be calculated because of missing data on surface heat fluxes (described in section 2.2.3) or other variables, and the omission of periods with missing relative humidity would have further decreased the number of calculated $k_{TE}$ values. Thus, we chose to approximate the missing values of relative humidity.

Furthermore, relative humidity has a very small effect on the gas transfer velocity calculated with the parameterization by Tedford et al. (2014) ($k_{TE}$), and it is not included in the other parameterizations. The Air-Sea Toolbox utilized by MyLake includes a formula that calculates air density $\rho_a$ on the basis of air temperature, relative humidity, and air pressure. The variation of air density at different relative humidities compared to the value of 70 % is, at maximum, of the order of 0.1 %. Because $k_{TE}$ is proportional to $\rho_a^{3/8}$, the corresponding variation of the gas transfer velocity is even smaller. However, we chose to include the statement on the relative humidity for completeness.

The mean value of the SMEAR measurements of relative humidity during the period May–October 2013 was 72 %. Platform measurements of relative humidity were relatively well applicable during the period May–August 2013. During May–August, the average values for the SMEAR and platform measurements of relative humidity were 66 % and 68 %, respectively. Thus, the average relative humidity can be assumed to be slightly higher over the lake than at the SMEAR station. Consequently, we find that a value of 75 % is a relatively good as a rough approximation for the average relative humidity at the platform during May–October. We have explained the choice in the text:

P9 L27–32: "In the calculation of the water-side friction velocity, missing relative humidities were replaced by a value of 75 %, which is close to the average of the SMEAR II measurements of relative humidity in May–October 2013, 72 %. The corresponding averages over the period May–August 2013, for which platform measurements were rather well applicable, were 66 % and 68 % for the SMEAR II and platform measurements, respectively. Thus, the relative

humidity can be assumed to have been slightly higher over the lake than at the SMEAR II station."

**- In section 2.2.4 : All the summary goodness-of-fit statistics (NS, B\*, URMSE'\*) can be displayed nicely in a target diagram. See Jolliff et al. 2009 "Summary diagrams for coupled hydrodynamic-ecosystem model skill assessment"**
All the statistics are presented in tables in the Supplement. Because the manuscript already contains a rather high amount of figures, we consider that additional target graphs would not provide much additional value to the manuscript.

Referee #2

**The manuscript "The Applicability and consequences of the integration of alternative models for CO2 transfer velocity into a process-based lake model" of Petri Kiuru, Anne Ojala, Ivan Mammarella, Jouni Heiskanen, Kukka-Maaria Erkkilä, Heli Miettinen, Timo Vesala, and Timo Huttula is a interessting scientifc report about the performance of different gas exchange models for simulations of CO2 fluxes between lakes and atmosphere. The article represents the high scientific expertise of the finish research community. No doubt, the authors did a grandiose job. In my understanding, the article can be accepted after two minor improvements.**

**(1) The authors wrote on page 7, line 4 that the lake has a maximum width of only 0.3km. This raises the question whether the footprint of the EC measurements is really representative for the lake-atmosphere exchange. How did the authors approximate the width of the parabolic footprint? And how did the authors consider transversal advection, i.e., advection orthogonally to the mean flow (wind) direction?**
The estimation of the flux footprint distribution functions was made using the model by Kormann and Meixner (2001). The average footprint contributing to 80 % of the flux ranges from 100 m up to about 300 m from the measurement platform depending on atmospheric stability conditions as described in Mammarella et al. (2015). However, the simple footprint model may have overestimated the footprint because it does not take into account the additional turbulence generated by the surrounding forest. Neverthless, it is justified to assume that the source area of the measured fluxes was on the lake surface because only the measurements during the periods when the wind was blowing along the lake were used in the analysis.
The wind is channeled along the lake for most of the time. When the wind is blowing along the lake, the footprints are within the lake fetch. Transversal wind directions were filtered out in the data used in the study. Typically, 15 % of the flux data are excluded from the analysis, when the wind is not blowing along the lake (the excluded wind directions are 350°–130° and 180°–320°). However, in calm nights some air can be transversally advected even if the wind is along the lake. In principle, the standard quality checking (described in detail in Mammarella et al. (2015)) removes the data contaminated by advection. Although the advection may still affect the concentrations and temperatures, the covariances with wind, that is, the eddy fluxes, are somewhat immune to advective effects.
We have added discussion on the aforementioned issues in section 2.2.3.
P9 L33–P10 L4: "The estimation of the flux footprint distribution functions was made using the model by Kormann and Meixner (2001). The average footprint contributing to 80 % of the fluxes varies from 100 m up to about 300 m from the measurement platform depending on atmospheric stability conditions as described in Mammarella et al. (2015). Only wind directions along the lake (130°–180°and 320°–350°) were included to ensure that heat fluxes from the surrounding land were excluded. Furthermore, possible remaining effects of transversal

advection during calm nights were removed through EC quality screening."

**(2) The authors discuss in section "4.2 Comparison to CO2 flux measurement" potential reasons for discrepancies between EC flux measurements and simulations results. Especially, they mentioned measurement errors and the spatial variability of governing parameters as major reasons. In my understanding, the authors are completely right with this statement. However, I would like to encourage the authors to provide quantitative support for this statement through a short error analysis.**

Estimates of the random uncertainty of EC fluxes on Lake Kuivajärvi for the years 2010 and 2011 have been studied in detail in Mammarella et al. (2015). On average, the estimated total relative random error was around 10 % for both sensible and latent heat fluxes. The estimated relative $CO_2$ flux random error was approximately double as large as that of energy fluxes, 20 % of measured fluxes, which is a typical value for EC $CO_2$ flux reported also in other types of ecosystems.

We have included some quantitative error analysis, related to both the underestimation of surface heat fluxes and the random measurement error, in the text:

P19 L26–31: "[...] The differences may be in part attributed to an underestimation of surface heat fluxes by the EC method, which was seen, for example, in a study on energy balance over a small boreal lake by Nordbo et al. (2011) and also in Mammarella et al. (2015). The sum of the measured EC heat fluxes in Lake Kuivajärvi was on average 83 % and 79 % of available energy in 2010 and 2011, respectively, in Mammarella et al. (2015). In addition, the total relative random error of the EC measurements is generally around 10 % for both sensible heat flux and latent heat flux as estimated in Mammarella et al. (2015). [...]"

**List of major changes made to the manuscript**

- A more detailed description of the applied gas transfer velocity parameterizations

  - A new, separate section (2.1.1) for presenting different ways of parameterization of gas exchange

  - A separate section (2.1.2) for presenting the applied models for the gas transfer velocity and their prior usage

- More detailed descriptions of the footprint of the EC fluxes (in section 2.2.3) and the error of EC heat flux measurements (in section 4.2)

- A new section (4.4) on modeling advice (recommendations on the selection of a gas exchange model on the basis of our intercomparison study).

- Conclusions: Highlighting the impact that the choice of more accurate gas exchange parametrizations has on global efflux from lakes

- A new table (Table 1) for the parameters used in the applied parameterizations of the gas transfer velocity

- Figures 3 and 5: Inclusion of cross plots that also contain some summary statistics along with each subplot for the gas transfer velocity (Figure 3) and for the $CO_2$ flux (Figure 5)

[revised manuscript text omitted]

---

## Author Response (AR2)

bg-2019-95
**Technical corrections**

We would like to thank the editor for the effort and time he put in to review our manuscript. We have made the required corrections to the corrected manuscript or otherwise clarified the issues that the editor had raised.

In the following list, the points raised by the editor are written in **bold** characters, whereas our responses are shown in blue characters and the text excerpts indicating the corresponding changes made to the manuscript are shown in orange characters. The line numbers within our responses correspond to those in the corrected manuscript file.

**Dear authors**
**thank you for the revision of your MS. I think your paper can be published in Biogeosciences after you make some minor technical corrections listed below.**
**Thank you for submitting your work to Biogeosciences**
**best regards,**
**Gwenaël Abril**

**P1L15 "However, finding higher estimates for both the internal and the external sources of inorganic carbon in boreal lakes is important if the improved knowledge of the magnitude of CO2 evasion from lakes is included in future studies on lake carbon budgets."**
**"finding higher estimate is important" do you mean that a CO2 source is actually missing?**
We mean that the estimates of internal and/or external sources of inorganic carbon may have been too low in many studies on lake carbon budgets because the air-lake $CO_2$ exchange calculations have been performed using simple and possible incorrect formulas/models. All relevant $CO_2$ sources may have been included in the studies, but they may have been too small. We see no reason to revise the sentence in the Abstract.

**P2L2 remove "found to be"**
Corrected.
P2 L2: "The majority of inland waters, especially in the boreal zone, are  supersaturated with carbon dioxide ($CO_2$) [. . . ]"

**P2L6 "but global quantitative estimates show significant variation", I have some doubt on the meaning of "variation" here. Do you mean that global estimates of lake CO2 emissions are uncertain?**
We meant that the estimates of the budget for the role of inland water ecosystems in the global carbon cycle show a lot variation, which further means that the exact quantitative contribution of lakes to the global carbon budget is uncertain. We have clarified the sentence and corrected the wording.
P2 L4-7: "The contribution of lakes to the global carbon budget is recognized to be substantial in comparison to the role of marine and terrestrial ecosystems as global carbon sinks, but quantitative estimates of the global contribution of lakes and other inland waters show significant variation (Cole et al., 2007; Battin et al., 2009; Tranvik et al., 2009)."

**"parameterized by wind speed" > "as a function of wind speed"?**
We have corrected the wording on P2 and a in similar occurrence on P4.
P2 L11-12: "In many long-used models for the gas transfer velocity, or gas exchange models, $k$ is parameterized as a function of wind speed alone [. . . ]"

P4L13: "The gas transfer velocity $k$ can be simply parameterized as a function of wind speed alone [. . . ]"

**P2L32 "revised estimates of lacustrine CO2 emissions will require higher terrestrial ecosystem production to close the global carbon balance" not clear; do you mean if k is higher, more C from terrestrial systems would be necessary to balance the lake C budget? Why higher terrestrial ecosystem production and not lower soil respiration and/or enhance erosion?**

In the so called "conventional carbon cycle", in which the three carbon resevoirs are land (terrestrial biosphere), the oceans (marine biosphere), and the atmosphere, inland waters are included in the terrestrial ecosystems or the terrestrial biosphere. 'Ecosystem production' stands for terrestrial net ecosystem production in the source that we used, Battin et al. (2009). Soil respiration and erosion are included in terrestrial net ecosystem production as a part of terrestrial ecosystem respiration in many models of global carbon cycling. Also, $CO_2$ efflux from inland waters is included in the fluxes of terrestrial ecosystem respiration, and secondary production and consequent respiration by heterotrophic biota in inland waters is not taken into account (Battin et al., 2009). If the separate carbon effluxes from inland waters are taken into account, net production in the land-based ecosystems of the terrestrial biosphere (which we referred to as terrestrial ecosystem production, following Battin et al. (2009)) will have to be increased. We have clarified the wording in the text.

P2 L30-32: "Thus, revised estimates of lacustrine $CO_2$ emissions will require higher net ecosystem production in the land-based ecosystems of the terrestrial biosphere to close the global carbon balance (Battin et al., 2009)."

**P6L9: "k is parameterized by the total turbulent kinetic energy dissipation rate" I am not sure "parameterized by" is correct wording**

Corrected.

P6 L9-10: "In the surface renewal model of air–water gas exchange, $k$ is parameterized as a function of the total turbulent kinetic energy dissipation rate [. . . ]"

**P6L15 add few lines on the main conclusions of the intercomparison study**

We have included some discussion on the results of the gas transfer velocity intercomparison study by Dugan et al. (2016) in section 4.1. We find that the Discussion section is a more appropriate location to shortly outline the results. In the Materials and Methods section, we have merely listed models and studies that use the gas exchange models according to the suggestions of Referee #1.

**P7L10 "the conversion of DOC into an inorganic form" > "the conversion of DOC into DIC"**

Corrected.

P7 L9-10: "A separate submodule (Holmberg et al., 2014) calculates the conversion of DOC into DIC via bacterial and photochemical degradation."

**P11L5 not clear what you mean by "the parameters related to interactions between DO and CO2"**

We have clarified the sentence.

P11 L5-6: "[. . . ] the parameters related to interactions between DO and $CO_2$, the photosynthetic quotient and the respiratory quotient, were excluded from the parameter set."

**P12L14 "because of system malfunction" not clear, do you mean the CO2 sensors are not**

**working?**

The possible reasons for the incorrect measurement results during the ice-covered period are difficult to specify. The measurement system consisted of gas-impermeable and semipermeable tubing, a diaphragm pump, and a $CO_2$ analyzer. Erroneous operation of any of the components may have caused the incorrect measurements. We have rephrased the sentence.

P12 L12-14: "However, the $CO_2$ concentration measurements performed during the ice-covered periods were [...] sometimes inapplicable also at deeper levels because of incorrect functioning of the measurement system."

**P12L18 "Only the days with applicable corresponding water column CO2 concentration measurement data were included in the averaging." What does "applicable" mean here, and what does "averaging" refer to? Average of what? please rephrase**

We have clarified and rephrased the sentence.

P12 L18-19: "Only the days with available corresponding water column $CO_2$ concentration measurement data were included in the averaging of the simulated near-surface concentrations over the open water seasons."

**L19 "the open water season averages of the measured near-surface (0.2 m) CO2 concentrations were..." awkward sentence, please rephrase**

We have rephrased the sentence.

P12 L19-20: "By contrast, the averages of the measured near-surface (0.2 m) $CO_2$ concentrations over the open water seasons were [...]"

**P20L6 "the different outcomes of the calibration processes can be considered equally applicable" it is hard to understand what you mean here, please explain**

We meant that it is justified to use the different lake model versions in the performance analysis and in the comparison of the model results with the measuremental results even though the parameters obtained in the calibrations were quite different between the lake model versions. The calibrations using different gas exchange models yielded rather differing parameter sets because of the random nature of the statistical calibration algorithm. However, each calibration of the lake model yielded a unique, possible description of the lake carbon dynamics, and the simulation results for $CO_2$ concentration and flux were rather similar between the different versions of the lake model. We have clarified the statement.

[revised manuscript text omitted]